# Counterfactual Conservative Q Learning for Offline Multi-agent Reinforcement Learning

**Jianzhun Shao**,* **Yun Qu**,* **Chen Chen, Hongchang Zhang, Xiangyang Ji**
Department of Automation
Tsinghua University, Beijing, China
{sjz18, qy22, hc-zhang19}@mails.tsinghua.edu.cn
cclvr@163.com
xyji@tsinghua.edu.cn

## Abstract

Offline multi-agent reinforcement learning is challenging due to the coupling effect of both distribution shift issue common in offline setting and the high dimension issue common in multi-agent setting, making the action out-of-distribution (OOD) and value overestimation phenomenon excessively severe. To mitigate this problem, we propose a novel multi-agent offline RL algorithm, named CounterFactual Conservative Q-Learning (CFCQL) to conduct conservative value estimation. Rather than regarding all the agents as a high dimensional single one and directly applying single agent methods to it, CFCQL calculates conservative regularization for each agent separately in a counterfactual way and then linearly combines them to realize an overall conservative value estimation. We prove that it still enjoys the underestimation property and the performance guarantee as those single agent conservative methods do, but the induced regularization and safe policy improvement bound are independent of the agent number, which is therefore theoretically superior to the direct treatment referred to above, especially when the agent number is large. We further conduct experiments on four environments including both discrete and continuous action settings on both existing and our man-made datasets, demonstrating that CFCQL outperforms existing methods on most datasets and even with a remarkable margin on some of them.

## 1 Introduction

Online Reinforcement Learning (Online RL) needs frequently deploying untested policies to environment for data collection and policy optimization, making it dangerous and inefficient to apply in the real-world scenarios (e.g. autonomous vehicle teams). While, Offline Reinforcement Learning (Offline RL) aims to learn policies from a fixed dataset rather than from interacting with the environment, and therefore is suitable for the real applications with highly safety requirements or without efficient simulators [25].

Directly applying off-policy RL to the offline setting may fail due to overestimation [13, 24]. Existing works usually tackle this problem by pessimism. They either utilize behavior regularization to constrain the learning policy close to the behavior policy induced by the dataset [49, 20, 11], or conduct conservative(pessimistic) value iteration to mitigate unexpected overestimation [18, 21]. It has been demonstrated both theoretically and empirically that these methods can achieve comparable performance to their online counterparts under some conditions [21].

---

*Equal contribution.

37th Conference on Neural Information Processing Systems (NeurIPS 2023).

Though cooperative Multi-agent Reinforcement Learning (MARL) has gained extraordinary success in various multiplayer video games such as Dota [4], StarCraft [40] and soccer [23], applying current MARL in real scenarios is still challenging due to the same safety and efficiency concerns in single-agent setting, then it is worth conducting investigation for offline RL in multi-agent setting. Compared with single-agent setting, offline RL in multi-agent setting has its own difficulty. On the one hand, the action space explodes exponentially as the agent number increases, then an arbitrary joint action is more likely to be an OOD action given the fixed dataset size, making the OOD phenomenon more severe. This will further exacerbate the issues of extrapolation error and overestimation in the policy evaluation process and thus induce an unexpected or even disastrous final policy [52]. On the other hand, as the core of MARL, the agents need to consider not only their own actions but also other agents' actions as well as contributions to the global return in order to achieve an overall high performance, which undoubtedly increases the difficulty for theoretical analysis. Besides, instead of just guaranteeing single policy improvement, the bounded team performance is also a key concern to offline MARL.

There exist few works to combine MARL with offline RL. Pan et al. [34] uses Independent Learning [44] as a simple solution, i.e., each agent regards others as part of the environment and independently performs offline RL learning. Although mitigating the joint action OOD issue through decoupling the agents's learning completely, it essentially still adopts a single-agent paradigm to learn and thus cannot enjoy the recent progress on MARL that a centralized value function can empower better team coordination [37, 27]. To utilize the advantage of centralized training with decentralized execution (CTDE), Yang et al. [52] applies implicit constraint approach on the value decomposition network [43], which alleviates the extrapolation error and gains some performance improvement empirically. Although it proposes some theoretical analysis for the convergence property of value function, whether the team performance can be bounded from below as agents number increases remains still unknown.

In this paper, we introduce a novel offline MARL algorithm called Counterfactual Conservative Q-Learning (CFCQL) and aim to address the overestimation issue rooted from joint action OOD phenomenon. It adopts CTDE paradigm to realize team coordination, and incorporates the state-of-the-art offline RL algorithm CQL to conduct conservative value estimation. CQL is preferred due to its theoretical guarantee and flexibility of implementation. One direct treatment is to regard all the agents as a single one and conduct standard CQL learning on the joint policy space which we call MACQL. However, too much conservatism will be generated in this way since the induced penalty can be exponentially large in the joint policy space. Instead, CFCQL separately regularizes each agent in a counterfactual way to avoid too much conservatism. Specifically, each agent separately contributes CQL regularization for the global Q value and then a weighted average is used as an overall regularization. When calculating agent $i$'s regularization term, rather than sampling OOD actions from the joint action space as MACQL does, we only sample OOD actions for agent $i$ and leave other agents' actions sampled in the dataset. We prove that CFCQL enjoys underestimation property as CQL does and the safe policy improvement guarantee independent of the agents number $n$, which is advantageous to MACQL especially under the situation with a large $n$.

We conduct experiments on 1 man-made environment Equal Line, and 3 commonly used multi-agent environments: StarCraft II [40], Multi-agent Particle Environment [27], and Multi-agent MuJoCo [35], including both discrete and continuous action space setting. With datasets collected by Pan et al. [34] and ourselves, our method outperforms existing methods in most settings and even with a large margin on some of them.

We summarize our contributions as follows: (1) we propose a novel offline MARL method CFCQL based on CTDE paradigm to address the overestimation issue and achieve team coordination at the same time. (2) We theoretically compare CFCQL and MACQL to show that CFCQL is advantagous to MACQL on the performance bounds and safe policy improvement guarantee as agent number is large. (3) In hard multi-agent offline tasks with both discrete and continuous action space, our method shows superior performance to the state-of-the-art.

## 2 Related Works

**Offline RL.** Standard RL algorithms are especially prone to fail due to erroneous value overestimation induced by the distributional shift between the dataset and the learning policy. Theoretically, it is

proved that pessimism can alleviate overestimation effectively and achieve good performance even with non-perfect data coverage [5, 15, 26, 22, 39, 51, 55, 6]. In the algorithmic line, there are broadly two categories: uncertainty based ones and behavior regularization based ones. Uncertainty based approaches attempt to estimate the epistemic uncertainty of Q-values or dynamics, and then utilize this uncertainty to pessimistically estimating Q in a model-free manner [2, 50, 3], or conduct learning on the pessimistic dynamic model in a model-based manner [54, 16]. Behavior regularization based algorithms constrain the learned policy to lie close to the behavior policy in either explicit or implicit ways [20, 13, 49, 18, 21, 11], and is advantageous over uncertainty based methods in computation efficiency and memory consumption. Among these class of algorithms, CQL[21] is preferred due to its superior empirical performance and flexibility of implementation.

**MARL.** A popular paradigm for multi-agent reinforcement learning is centralized training with decentralized execution (CTDE) [37, 27]. Centralized training inherits the idea of joint action learning [7], empowering the agents with better team coordination. And decentralized execution enjoys the deploying flexibility of independent learning [44]. In CTDE, some value-based works concentrate on decomposing the single team reward to all agents by value function factorization [43, 37], based on which they further derive and extend the Individual-Global-Max (IGM) principle for policy optimality analysis [42, 47, 38, 46]. Another group of works focus on the actor-critic framework, using a centralized critic to guide each agent's policy update [27, 9]. Some variants of PPO [41], including IPPO [8], MAPPO [53], and HAPPO [19] also show great potential in solving complex tasks. All these works use the online learning paradigm, therefore disturbed by extrapolation error when transferred to the offline setting.

**Offline MARL.** To make MARL applied in the more practical scenario with safety and training efficiency concerns, offline MARL is proposed. Jiang & Lu [14] shows the challenge of applying BCQ [13] to independent learning, and Pan et al. [34] uses zeroth-order optimization for better coordination among agents' policies. Although it empirically shows fast convergence rate, the policies trained by independent learning have no theoretical guarantee for the team performance. With CTDE paradigm, Yang et al. [52] adopts the same treatment as Peng et al. [36] and Nair et al. [30] to avoid sampling new actions from current policy, and Tseng et al. [45] regards the offline MARL as a sequence modeling problem, solving it by supervised learning. As a result, both methods' performance relies heavily on the data quality. In contrast, CFCQL does not require the learning policies stay close to the behavior policy, and therefore performs well on datasets with low quality. Meanwhile, CFCQL is theoretically guaranteed to be safely improved, which ensures its performance on datasets with high quality.

## 3 Preliminary

### 3.1 MARL Symbols

We use a *decentralised partially observable Markov decision process* (Dec-POMDP) [32] $G = \langle S, \mathbf{A}, I, P, r, Z, O, n, \gamma \rangle$ to model a fully collaborative multi-agent task with $n$ agents, where $s \in S$ is the global state. At time step $t$, each agent $i \in I \equiv \{1, ..., n\}$ chooses an action $a^i \in A$, forming the joint action $\mathbf{a} \in \mathbf{A} \equiv A^n$. $T(s'|s, \mathbf{a}) : S \times \mathbf{A} \times S \to [0, 1]$ is the environment's state transition distribution. All agents share the same reward function $r(s, \mathbf{a}) : S \times \mathbf{A} \to \mathbb{R}$. $\gamma \in [0, 1)$ is the discount factor. Each agent $i$ has its local observations $o^i \in O$ drawn from the observation function $Z(s, i) : S \times I \to O$ and chooses an action by its policy $\pi^i(a^i|o^i) : O \to \Delta([0, 1]^{|A|})$. The agents' joint policy $\boldsymbol{\pi} := \prod_{i=1}^{n} \pi^i$ induces a joint *action-value function*: $Q^{\boldsymbol{\pi}}(s, \mathbf{a}) = \mathbb{E}[R|s, \mathbf{a}]$, where $R = \sum_{t=0}^{\infty} \gamma^t r_t$ is the discounted accumulated team reward. We assume $\forall r, |r| \leq R_{\max}$. The goal of MARL is to find the optimal joint policy $\boldsymbol{\pi}^*$ such that $Q^{\boldsymbol{\pi}^*}(s, \mathbf{a}) \geq Q^{\boldsymbol{\pi}}(s, \mathbf{a})$, for all $\boldsymbol{\pi}$ and $(s, \mathbf{a}) \in S \times \mathbf{A}$. In offline MARL, we need to learn the optimal $\boldsymbol{\pi}^*$ from a fixed dataset sampled by an unknown behaviour policy $\boldsymbol{\beta}$, and we can not deploy any new policy to the environment to get feedback.

### 3.2 Value Functions in MARL

In MARL settings with discrete action space, we can maintain for each agent a local-observation-based Q function $Q^i(o^i, a^i)$, and define the local policy $\pi^i := \arg\max_a Q^i(o^i, a)$. To train local policies with single team reward, Rashid et al. [37] proposes QMIX, using a global $Q^{tot}$ to model

the cumulative team reward. Define the Bellman operator $\mathcal{T}$ as $\mathcal{T}Q(s, \boldsymbol{a}) = r + \gamma \max_{\boldsymbol{a}'} Q(s', \boldsymbol{a}')$. QMIX aims to minimize the temporal difference:

$$\hat{Q}_{k+1}^{tot} \leftarrow \arg \min_Q \mathbb{E}_{s, \boldsymbol{a}, s' \sim \mathcal{D}}[(Q(s, \boldsymbol{a}) - \hat{\mathcal{T}}\hat{Q}_k^{tot}(s, \boldsymbol{a}))^2], \tag{1}$$

where $\hat{\mathcal{T}}$ is the empirical Bellman operator using samples from the replay buffer $\mathcal{D}$, and $\hat{Q}^{tot}$ is the empirical global Q function, computed from $s$ and $Q^i$s represented by a neural network satisfying $\partial Q^{tot} / \partial Q^i \geq 0$. With such restriction, QMIX can ensure the Individual-Global-Max principle that $\arg \max_{\boldsymbol{a}} Q^{tot}(s, \boldsymbol{a}) = (\arg \max_{a^1} Q^1(o^1, a^1), ..., \arg \max_{a^n} Q^n(o^n, a^n))$. For continuous action space, a neural network is used to represent the local policy $\pi^i$ for each agent. Lowe et al. [27] proposes MADDPG, maintaining a centralized critic $Q^{tot}$ to directly guide the local policy update by gradient descent. And the update rule of $Q^{tot}$ is similar to Eq. 1, just replacing $\mathcal{T}$ with $\mathcal{T}^{\boldsymbol{\pi}}$. $\mathcal{T}^{\boldsymbol{\pi}}Q = r + \gamma P^{\boldsymbol{\pi}}Q$, where $P^{\boldsymbol{\pi}}Q(s, \boldsymbol{a}) = \mathbb{E}_{s' \sim T(\cdot|s, \boldsymbol{a}), \boldsymbol{a}' \sim \boldsymbol{\pi}(\cdot|s')}[Q(s', \boldsymbol{a}')]$.

### 3.3 Conservative Q-Learning

Offline RL is prone to the overestimation of value functions. For an intuitive explanation of the underlying cause, please refer to Appendix E.1. Kumar et al. [21] proposes Conservative Q-Learning (CQL), adding a regularizer to the Q function to address the overestimation issue, which maximizes the Q function of the state-action pairs from the dataset distribution $\beta$, while penalizing the Q function sampled from a new distribution $\mu$ (e.g., current policy $\pi$). The conservative policy evaluation is shown as follows:

$$\hat{Q}_{k+1} \leftarrow \arg \min_Q \alpha[\mathbb{E}_{s \sim \mathcal{D}, a \sim \mu}[Q(s, a)] - \mathbb{E}_{s \sim \mathcal{D}, a \sim \beta}[Q(s, a)]] + \frac{1}{2}\mathbb{E}_{s, a, s' \sim \mathcal{D}}[(Q(s, a) - \hat{\mathcal{T}}^\pi \hat{Q}_k(s, a))^2]. \tag{2}$$

As shown in Theorem 3.2 in Kumar et al. [21], with a large enough $\alpha$, we can obtain a Q function, whose expectation over actions is lower than that of the real Q, then the overestimation issue can be mitigated.

## 4 Proposed Method

We first show the overestimation problem of multi-agent value functions in Sec. 4.1, and a direct solution by the naive extension of CQL to multi-agent settings in Sec. 4.2, which we call MACQL. In Sec. 4.3, we propose Counterfactual Conservative Q-Learning (CFCQL) and further demonstrate that it can realize conservatism in a more mild and controllable way which is independent of the number of agents. Then we compare MACQL and CFCQL from the aspect of safe improvement performance. In Sec. 4.4, we present our novel and practical offline RL algorithm.

### 4.1 Overestimation in Offline MARL

The Q-values in online learning usually face the overestimation problem [1, 33], which becomes more severe in offline settings due to distribution shift and offline policy evaluation [13]. Offline MARL suffers from the same issue even worse since the joint action space explodes exponentially as the agent number increases, then an arbitrary joint action is more likely to be an OOD action given the fixed dataset size. To illustrate this phenomenon, we design a toy Multi-Agent Markov Decision Process (MMDP) as shown in Fig.1(a). There are five agents and three states. All agents are randomly initialized and need learn to take three actions

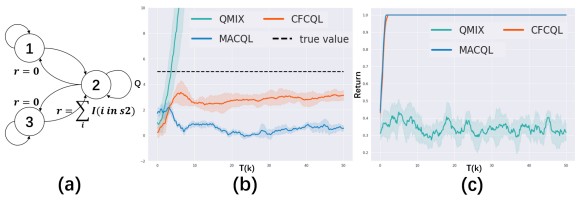

(a)        (b)        (c)

Figure 1: (a) A decomposable Multi-Agent MDP which urges the agents moves to or stays in $S2$. The number of agents is set to 5. (b) The learning curve of the estimated joint state-action value function in the given MMDP. The true value is indicated by the dotted line, which represents the maximum discounted return. (c) The performance curve of the corresponding methods.

(stay, move up, move down) to move to or stay in the state $S2$. The dataset consists of 1000 samples with reward larger than $0.8r_{\max}$. As shown in Fig. 1(b), if not specially designed for offline setting,

naive application of QMIX leads to exponentially explosion of value estimates and the sub-optimal policy performance, Fig. 1(c).

## 4.2 Multi-Agent Conservative Q-Learning

When treating multiple agents as a unified single agent and only regarding the joint policy of the $\boldsymbol{\beta}, \boldsymbol{\mu}, \boldsymbol{\pi}$, that is $\boldsymbol{\beta}(\boldsymbol{a}|s) = \prod_{i=1}^{n} \beta^i(a^i|s)$, $\boldsymbol{\mu}(\boldsymbol{a}|s) = \prod_{i=1}^{n} \mu^i(a^i|s)$ and $\boldsymbol{\pi}(\boldsymbol{a}|s) = \prod_{i=1}^{n} \pi^i(a^i|s)$, we can extend the single-agent CQL to CTDE multi-agent setting straightforwardly and derive the conservative policy evaluation style directly as follows:

$$\hat{Q}_{k+1}^{tot} \leftarrow \arg\min_Q \alpha \left[ \mathbb{E}_{s\sim\mathcal{D}, \boldsymbol{a}\sim\boldsymbol{\mu}}[Q(s,\boldsymbol{a})] - \mathbb{E}_{s\sim\mathcal{D}, \boldsymbol{a}\sim\boldsymbol{\beta}}[Q(s,\boldsymbol{a})] \right] + \hat{\mathcal{E}}_{\mathcal{D}}(\boldsymbol{\pi}, Q, k), \quad (3)$$

where we represent the temporal difference (TD) error concisely as $\hat{\mathcal{E}}_{\mathcal{D}}(\boldsymbol{\pi}, Q, k) := \frac{1}{2}\mathbb{E}_{s,\boldsymbol{a},s'\sim\mathcal{D}}[(Q(s,\boldsymbol{a}) - \hat{\mathcal{T}}^{\boldsymbol{\pi}}\hat{Q}_k^{tot}(s,\boldsymbol{a}))^2]$. In the rest of the paper, we mainly focus on the global Q function of a centralized critic and omit the superscript "tot" for ease of expression.

Similar to CQL, MACQL can learn a lower-bounded Q-value with a proper $\alpha$, then the overestimation issue in offline MARL can be mitigated. According to Theorem 3.2 in Kumar et al. [21], the degree of pessimism relies heavily on $D_{CQL}(\boldsymbol{\pi}, \boldsymbol{\beta})(s) := \sum_{\boldsymbol{a}} \boldsymbol{\pi}(\boldsymbol{a}|s)[\frac{\boldsymbol{\pi}(\boldsymbol{a}|s)}{\boldsymbol{\beta}(\boldsymbol{a}|s)} - 1]$. However, as we will show in Sec. 4.3, $D_{CQL}$ expands exponentially as the number of agents $n$ increases, resulting in an over-pessimistic value function and a mediocre policy improvement guarantee for MACQL.

## 4.3 Counterfactual Conservative Q-Learning

Intuitively, we aim to learn a value function that prevents the overestimation of the policy value, which is expected not too far away from the true value in the meantime. Similar to MACQL as introduced above, we add a regularization in the policy evaluation process, by penalizing the values for OOD actions and rewarding the values for in-distribution actions. However, rather than regarding all the agents as a single one and conducting standard CQL method in the joint policy space, in CFCQL method, each agent separately contributes regularization for the global Q value and then a weighted average is used as an overall regularization. When calculating agent $i$'s regularization term, instead of sampling OOD actions from the joint action space as MACQL does, we only sample OOD actions for agent $i$ and leave other agents' actions sampled in the dataset. Specifically, using $\boldsymbol{a}^{-i}$ to denote the joint actions other than agent $i$. Eq. 4 is our proposed policy evaluation iteration:

$$\hat{Q}_{k+1} \leftarrow \arg\min_Q \alpha \left[ \sum_{i=1}^{n} \lambda_i \mathbb{E}_{s\sim\mathcal{D}, a^i\sim\mu^i, \boldsymbol{a}^{-i}\sim\boldsymbol{\beta}^{-i}}[Q(s,\boldsymbol{a})] - \mathbb{E}_{s\sim\mathcal{D}, \boldsymbol{a}\sim\boldsymbol{\beta}}[Q(s,\boldsymbol{a})] \right] + \hat{\mathcal{E}}_{\mathcal{D}}(\boldsymbol{\pi}, Q, k), \quad (4)$$

where $\alpha, \lambda_i$ are hyper-parameters, and $\sum_{i=1}^{n} \lambda_i = 1, \lambda_i \geq 0$. We refer to our method as 'counterfactual' because its structure bears resemblance to counterfactual methods in MARL [9]. This involves obtaining each agent's counterfactual baseline by marginalizing out a single agent's action while keeping the other agents' actions fixed. The intuitive rationale behind employing a counterfactual-like approach is that by individually penalizing each agent's out-of-distribution (OOD) actions while holding the others' actions constant from the datasets, we can effectively mitigate the out-of-distribution problem in offline MARL with reduced pessimism, as illustrated in the rest of this section.

The theoretical analysis is arranged as follows: In Theorem 4.1 we show the new policy evaluation leads to a milder conservatism on value function, which still lower bounds the true value. Then we compare the conservatism degree between CFCQL and MACQL in Theorem 4.2. In Theorem 4.3 and Theorem 4.4 we show the effect of milder conservatism brought by counterfactual treatments on the performance guarantee (All proofs are defered to Appendix A).

**Theorem 4.1** (Equation 4 results in a lower bound of value function). *The value of the policy under the Q function from Equation 4, $\hat{V}^{\boldsymbol{\pi}}(s) = \mathbb{E}_{\boldsymbol{\pi}(\boldsymbol{a}|s)}[\hat{Q}^{\boldsymbol{\pi}}(s,\boldsymbol{a})]$, lower-bounds the true value of the policy obtained via exact policy evaluation, $V^{\boldsymbol{\pi}}(s) = \mathbb{E}_{\boldsymbol{\pi}(\boldsymbol{a}|s)}[Q^{\boldsymbol{\pi}}(s,\boldsymbol{a})]$, when $\boldsymbol{\mu} = \boldsymbol{\pi}$, according to: $\forall s \in$*

$$\mathcal{D}, \hat{V}^{\boldsymbol{\pi}}(s) \leq V^{\boldsymbol{\pi}}(s) - \alpha \left[ (I - \gamma P^{\boldsymbol{\pi}})^{-1} \mathbb{E}_{\boldsymbol{\pi}} \left[ \sum_{i=1}^{n} \lambda_i \frac{\pi^i}{\beta^i} - 1 \right] \right](s) + \left[ (I - \gamma P^{\boldsymbol{\pi}})^{-1} \frac{C_{r,T,\delta} R_{\max}}{(1-\gamma)\sqrt{|\mathcal{D}|}} \right].$$

*Define $D_{CQL}^{CF}(\boldsymbol{\pi}, \boldsymbol{\beta})(s) := \sum_{\boldsymbol{a}} \boldsymbol{\pi}(\boldsymbol{a}|s)[\sum_{i=1}^{n} \lambda_i \frac{\pi^i(a^i|s)}{\beta^i(a^i|s)} - 1]$. If $\alpha > \frac{C_{r,T,\delta} R_{\max}}{1-\gamma} \cdot \max_{s\in\mathcal{D}} \frac{1}{\sqrt{|\mathcal{D}(s)|}} \cdot$*

$D_{CQL}^{CF}(\boldsymbol{\pi}, \boldsymbol{\beta})(s)^{-1}, \forall s \in \mathcal{D}, \hat{V}^{\boldsymbol{\pi}}(s) \leq V^{\boldsymbol{\pi}}(s)$, *with probability* $\geq 1 - \delta$. *When* $\mathcal{T}^{\boldsymbol{\pi}} = \hat{\mathcal{T}}^{\boldsymbol{\pi}}$, *then any* $\alpha > 0$ *guarantees* $\hat{V}^{\boldsymbol{\pi}}(s) \leq V^{\boldsymbol{\pi}}(s), \forall s \in \mathcal{D}$.

Theorem 4.1 mainly differs from Theorem 3.2 in Kumar et al. [21] on the specific form of the divergence: $D_{CQL}$ and $D_{CQL}^{CF}$, both of which determine the degrees of conservatism when $\boldsymbol{\pi}$ and $\boldsymbol{\beta}$ are different. Empirically, we find $D_{CQL}$ generally becomes too large when the number of agents expands, resulting in a low, even negative value function. To measure the scale difference between $D_{CQL}(\boldsymbol{\pi}, \boldsymbol{\beta})(s) := \prod_i(D_{CQL}(\pi^i, \beta^i)(s) + 1) - 1$ and $D_{CQL}^{CF}(\boldsymbol{\pi}, \boldsymbol{\beta})(s) := \sum_i \lambda_i D_{CQL}(\pi^i, \beta^i)(s)$, we have Theorem 4.2:

**Theorem 4.2.** $\forall s, \boldsymbol{\pi}, \boldsymbol{\beta}$, *one has* $0 \leq D_{CQL}^{CF}(\boldsymbol{\pi}, \boldsymbol{\beta})(s) \leq D_{CQL}(\boldsymbol{\pi}, \boldsymbol{\beta})(s)$, *and the following inequality holds:*

$$\frac{D_{CQL}(\boldsymbol{\pi}, \boldsymbol{\beta})(s)}{D_{CQL}^{CF}(\boldsymbol{\pi}, \boldsymbol{\beta})(s)} \geq \exp\left(\sum_{i=1, i \neq j}^n KL(\pi^i(s) || \beta^i(s))\right), \tag{5}$$

*where* $j = \arg\max_k \mathbb{E}_{\pi^k} \frac{\pi^k(s)}{\beta^k(s)}$ *represents the agent whose policy distribution is the most far away from the dataset.*

Since $D_{CQL}^{CF}$ can be written as $\sum_i \lambda_i D_{CQL}(\pi^i, \beta^i)(s)$, it can be regarded as a weighted average of each individual agents' policy deviations from its individual behavior policy. Therefore, the scale of $D_{CQL}^{CF}$ is independent of the number of agents. Instead, Theorem 4.2 shows that both $D_{CQL}$ and the ratio of $D_{CQL}$ and $D_{CQL}^{CF}$ explode exponentially as the number of agents $n$ increases.

Then we discuss the influence of $D_{CQL}^{CF}$ on the property of safe policy improvement. Define $\hat{M}$ as the MDP induced by the transitions observed in the dataset. Let the empirical return $J(\boldsymbol{\pi}, \hat{M})$ be the discounted return for any policy $\boldsymbol{\pi}$ in $\hat{M}$ and $\hat{Q}$ be the fixed point of Equation 4. Then the optimal policy of CFCQL is equivalently obtained by solving:

$$\boldsymbol{\pi}_{CF}^*(\boldsymbol{a}|s) \leftarrow \arg\max_{\boldsymbol{\pi}} J(\boldsymbol{\pi}, \hat{M}) - \alpha \frac{1}{1 - \gamma} \mathbb{E}_{s \sim d_{\hat{M}}^{\boldsymbol{\pi}}(s)}[D_{CQL}^{CF}(\boldsymbol{\pi}, \boldsymbol{\beta})(s)]. \tag{6}$$

See the proof in Appendix A.3. Eq. 6 can be regarded as adding a $D_{CQL}^{CF}$-related regularizer to $J(\boldsymbol{\pi}, \hat{M})$. Replacing $D_{CQL}^{CF}$ with $D_{CQL}$ in the regularizer, we can get a similar form of $\boldsymbol{\pi}_{MA}^*$ for MACQL. We next discuss the performance bound of CFCQL and MACQL on the true MDP $M$. If we assume the full coverage of $\beta^i$ on $M$, we have:

**Theorem 4.3.** *Assume* $\forall s, a, i, \beta^i(a|s) \geq \epsilon$, *then with probability* $\geq 1 - \delta$,

$$J(\boldsymbol{\pi}_{MA}^*, M) \geq J(\boldsymbol{\pi}^*, M) - \frac{\alpha}{1 - \gamma}(\frac{1}{\epsilon^n} - 1) - sampling\ error,$$

$$J(\boldsymbol{\pi}_{CF}^*, M) \geq J(\boldsymbol{\pi}^*, M) - \frac{\alpha}{1 - \gamma}(\frac{1}{\epsilon} - 1) - sampling\ error,$$

*where* $\boldsymbol{\pi}^* = \arg\max_{\boldsymbol{\pi}} J(\boldsymbol{\pi}, M)$, *i.e. the optimal policy, and sampling error is a constant dependent on the MDP itself and* $\mathcal{D}$.

Theorem 4.3 shows that when sampling error and $\alpha$ are small enough, the performances gap induced by $\boldsymbol{\pi}_{CF}^*$ can be small, but that induced by $\boldsymbol{\pi}_{MA}^*$ expands when $n$ increases, making $\boldsymbol{\pi}_{MA}^*$ far away from the optimal. Upon examining the performance gap, we ultimately compare the safe policy improvement guarantees of the two methods on $M$:

**Theorem 4.4.** *Assume* $\forall s, a, i, \beta^i(a|s) \geq \epsilon$. *The policy* $\boldsymbol{\pi}_{MA}^*(\boldsymbol{a}|s)$ *is a* $\zeta^{MA}$-*safe policy improvement over* $\boldsymbol{\beta}$ *in the actual MDP* $M$, *i.e.,* $J(\boldsymbol{\pi}_{MA}^*, M) \geq J(\boldsymbol{\beta}, M) - \zeta^{MA}$. *And the policy* $\boldsymbol{\pi}_{CF}^*(\boldsymbol{a}|s)$ *is a* $\zeta^{CF}$-*safe policy improvement over* $\boldsymbol{\beta}$ *in* $M$, *i.e.,* $J(\boldsymbol{\pi}_{CF}^*, M) \geq J(\boldsymbol{\beta}, M) - \zeta^{CF}$. *When* $n \geq \log_{\frac{1}{\epsilon}}\left(\frac{1}{\epsilon} + \frac{2}{\alpha} \frac{\sqrt{|A|}}{\sqrt{|\mathcal{D}(s)|}}(C_{r,\delta} + \frac{\gamma R_{\max} C_{T,\delta}}{1-\gamma}) \cdot (\frac{1}{\sqrt{\epsilon}} - 1)\right), \zeta^{CF} \leq \zeta^{MA}$.

Detailed formation of $\zeta^{MA}$ and $\zeta^{CF}$ is provided in Appendix A.5. Theorem 4.4 shows that with a large enough agent count $n$, CFCQL has better safe policy improvement guarantee than MACQL. The validation experiment of this theoretical result is presented in Section 5.1.

## 4.4 Practical Algorithm

The fixed point of Eq. 4 provides an underestimated Q function for any policy $\mu$. But it is computationally inefficient to solve Eq. 4 every time after one step policy update. Similar to CQL [21], we also choose a $\mu(\boldsymbol{a}|s)$ that would maximize the current $\hat{Q}$ with a $\mu$-regularizer. If the regularizer aims to minimize the KL divergence between $\mu^i$ and a uniform distribution, $\mu^i(a^i|s) \propto \exp\left(\mathbb{E}_{\boldsymbol{a}^{-i} \sim \boldsymbol{\beta}^{-i}} Q(s, a^i, \boldsymbol{a}^{-i})\right)$, which results in the update rule of Eq. 7 (See Appendix B.1 for detailed derivation):

$$\min_Q \alpha \mathbb{E}_{s \sim \mathcal{D}}\left[\sum_{i=1}^n \lambda_i \mathbb{E}_{\boldsymbol{a}^{-i} \sim \boldsymbol{\beta}^{-i}}[\log \sum_{a^i} \exp(Q(s, \boldsymbol{a}))] - \mathbb{E}_{\boldsymbol{a} \sim \boldsymbol{\beta}}[Q(s, \boldsymbol{a})]\right] + \hat{\mathcal{E}}_{\mathcal{D}}(\boldsymbol{\pi}, Q). \quad (7)$$

Finally, we need to specify each agent's weight of minimizing the policy Q function $\lambda_i$. Theoretically, any simplex of $\boldsymbol{\lambda}$ that satisfies $\sum_{i=1}^n \lambda_i = 1$ can be used to induce an underestimated value function linearly increasing as agents number as we expect. Therefore, a simple way is to set $\lambda_i = \frac{1}{n}, \forall i$ where each agent contributes penalty equally. Another way is to prioritize penalizing the agent that exhibits the greatest deviation from the dataset, which is the one-hot style of $\boldsymbol{\lambda}$:

$$\lambda_i(s) = \begin{cases} 1.0, & i = \arg\max_j \mathbb{E}_{\pi^j} \frac{\pi^j(s)}{\beta^j(s)} \\ 0.0, & others \end{cases} \quad (8)$$

We assert that each agent's conservative contribution deserves to be considered and differed according to their degree of deviation. As a result, both the uniform and the one-hot treatment present some limitations. Consequently, we employ a moderate softmax variant of Eq. 8:

$$\forall i, s, \lambda_i(s) = \frac{\exp(\tau \mathbb{E}_{\pi^i} \frac{\pi^i(s)}{\beta^i(s)})}{\sum_{j=1}^n \exp(\tau \mathbb{E}_{\pi^j} \frac{\pi^j(s)}{\beta^j(s)})}, \quad (9)$$

where $\tau$ is a predefined temperature coefficient, controlling the influence of $\mathbb{E}_{\pi^i} \frac{\pi^i}{\beta^i}$ on $\lambda_i$. When $\tau \to 0$, $\lambda_i \to \frac{1}{n}$, and when $\tau \to \infty$, it turns into Eq. 8. To compute Eq. 9, we need an explicit expression of $\pi^i$ and $\beta^i$. In discrete action space, $\pi^i$ can be estimated by $\exp\left(\mathbb{E}_{\boldsymbol{a}^{-i} \sim \boldsymbol{\beta}^{-i}} Q(s, a^i, \boldsymbol{a}^{-i})\right)$, and we use behavior cloning [29] to train a parameterized $\beta(s)$ from the dataset. In continuous action space, $\pi^i$ is parameterized by each agent's local policy. For $\beta^i$, we use the method of explicit estimation of behavior density in Wu et al. [48], which is modified from a VAE [17] estimator. Details for computing $\boldsymbol{\lambda}$ are defered to Appendix B.2.

For policy improvement in continuous action space, we also take derivation of a counterfactual Q function for each agent, rather than updating all agents' policy together like in MAD-DPG. Specifically, the gradient of each agent $i$'s policy $\pi^i$ is calculated by:

$$\nabla_{a^i} \mathbb{E}_{s, \boldsymbol{a}^{-i} \sim \mathcal{D}, a^i \sim \pi^i(s)} Q_\theta(s, \boldsymbol{a}) \quad (10)$$

The reason is that in CFCQL, we only minimize $Q(s, \pi^i, \boldsymbol{\beta}^{-i})$, rather than $Q(s, \boldsymbol{\pi})$. Using the untrained $Q(s, \boldsymbol{\pi})$ to directly guide PI like MADDPG may result in a bad policy.

We summarize CFCQL in discrete and continuous action space in Algorithm 1 as CFCQL-D and -C, separately.

---

**Algorithm 1** CFCQL-D and CFCQL-C

1: Initialize $Q_\theta$, target network $Q_{\hat{\theta}}$, target update interval $t_{tar}$, replay buffer $\mathcal{D}$, and optionally $\boldsymbol{\pi}_\psi$ for CFCQL-C
2: **for** $t = 1, 2, \ldots, t_{max}$ **do**
3:     Sample $N$ transitions $\{s, \boldsymbol{a}, s', r\}$ from $\mathcal{D}$
4:     Compute $Q_\theta(s, \boldsymbol{a})$ using the structure of QMIX for CFCQL-D or MADDPG for CFCQL-C.
5:     Calculate $\boldsymbol{\lambda}$ according to Eq. 9
6:     Update $Q_\theta$ by Eq. 7 with sampled transitions. Using $\hat{\mathcal{T}}_{\hat{\theta}}$ for CFCQL-D, and $\hat{\mathcal{T}}_{\hat{\theta}}^{\boldsymbol{\pi}_{\psi_t}}$ for CFCQL-C
7:     (Only for CFCQL-C) For each agent $i$, take one-step policy improvement for $\pi_\psi^i$ according to Eq. 10
8:     **if** $t \mod t_{tar} = 0$ **then**
9:         update target network $\hat{\theta} \leftarrow \theta$
10:     **end if**
11: **end for**

---

# 5 Experiments

**Baselines.** We compare our method CFCQL with several offline Multi-Agent baselines, where baselines with prefix $MA$ adopt CTDE paradigm and the others adopt independent learning paradigm: **BC**: Behavior cloning. **TD3-BC**[11]: One of the state-of-the-art single agent offline algorithm, simply adding the BC term to TD3 [12]. **MACQL**: Naive extension of conservative Q-learning, as proposed in Sec.4.2 . **MAICQ**[52]: Multi-agent version of implicit constraint Q-learning by decomposed multi-agent joint-policy under implicit constraint. **OMAR**[34]: Using zeroth-order optimization for better coordination among agents' policies, based on independent CQL (**ICQL**). **MADTKD**[45]: Using decision transformer to represent each agent's policy, trained with knowledge distillation. **IQL**[18] and **AWAC**[31]: variants of advantage weighted behaviour cloning, which are SOTA on single agent offline RL. Details for baseline implementations are in Appendix C.3.

Each algorithm is run for five random seeds, and we report the mean performance with standard deviation[2]. Four environments are adopted to evaluate our method, including both discrete action space and continuous action space. We have relocated additional experimental results to Appendix D to conserve space.

Figure 2: (a) The $Equal\_Line$ environment where n=3. (b) Performance ratio of CFCQL and MACQL to the behaviour policy with a varing number of agents.

## 5.1 Equal Line

To empirically compare CFCQL and MACQL as agents number $n$ increases, we design a multi-agent task called $Equal\_Line$, which is a one-dimensional simplified version of $Equal\_Space$ introduced in Tseng et al. [45]. Details of the environment are in Appendix C.1. The $n$ agents need to cooperate to disperse and ensure every agent is equally spaced. The datasets consist of 1000 trajectories sampled by executing a fully-pretrained policy of QMIX[37], i.e. $Expert$ dataset. We plot the performance ratios of CFCQL and MACQL to the behavior policy for different agent number $n$ in Fig.2(b). It can be observed that the performance of MACQL degrades dramatically as the increase of number of agents while the performance of CFCQL remains basically stable. The results strongly verify the conclusion we proposed in Sec.4.3, that CFCQL has better policy improvement guarantee than MACQL with a large enough number of agents $n$.

## 5.2 Multi-agent Particle Environment

In this section, we test CFCQL on Multi-agent Particle Environemnt with continuous action space. We use the dataset and the adversary agent provided by Pan et al. [34]. The performance of the trained model is measured by the normalized score $100 \times (S - S_{Random})/(S - S_{Expert})$ [10].

In Table 1, we only show the comparison of our method and the current state-of-the-art method OMAR and IQL to save space. For complete comparison with more baselines, e.g., TD3+BC and

Table 1: Results on Multi-agent Particle Environment. CN: Cooperative Navigation. PP:Predator-prey. World: World.

| Env | Dataset | OMAR | MACQL | IQL | CFCQL |
|---|---|---|---|---|---|
| CN | Random | 34.4±5.3 | 45.6±8.7 | 5.5±1.1 | **62.2±8.1** |
| | Med-Rep | 37.9±12.3 | 25.5±5.9 | 10.8±4.5 | **52.2±9.6** |
| | Medium | 47.9±18.9 | 14.3±20.2 | 28.2±3.9 | **65.0±10.2** |
| | Expert | **114.9±2.6** | 12.2±31 | 103.7±2.5 | 112±4 |
| PP | Random | 11.1±2.8 | 25.2±11.5 | 1.3±1.6 | **78.5±15.6** |
| | Med-Rep | 47.1±15.3 | 11.9±9.2 | 23.2±12 | **71.1±6** |
| | Medium | 66.7±23.2 | 55±43.2 | 53.6±19.9 | **68.5±21.8** |
| | Expert | 116.2±19.8 | 108.4±21.5 | 109.3±10.1 | **118.2±13.1** |
| World | Random | 5.9±5.2 | 11.7±11 | 2.9±4.0 | **68±20.8** |
| | Med-Rep | 42.9±19.5 | 13.2±16.2 | 41.5±9.5 | **73.4±23.2** |
| | Medium | 74.6±11.5 | 67.4±48.4 | 70.5±15.3 | **93.8±31.8** |
| | Expert | 110.4±25.7 | 99.7±31 | 107.8±17.7 | **119.7±26.4** |

---

[2]Our code and datasets are available at: `https://github.com/thu-rllab/CFCQL`

AWAC, please refer to Appendx D.1. It can be seen that on 11 of the 12 datasets, CFCQL shows superior performance than current state-of-the-art. Note that we only report the results of $\tau = 0$. In Appendix D.2 of the ablation on $\tau$, we show that with carefully fine-tuned $\tau$, higher scores of CFCQL can be obtained. We also carry out ablations of $\alpha$ in Appendix D.3.

## 5.3   Multi-agent MuJoCo

In this section we investigate the effect of our method on more complex continuous control task. We use the HalfCheetah-v2 setting from the multi-agent MuJoCo environment [35] and the datasets provided by Pan et al. [34]. Table 2 shows that CFCQL exceeds the current state-of-the-art on most datasets.

Table 2: Results on MaMuJoCo.

| Dataset | Random | Med-rep | Medium | Expert |
|---|---|---|---|---|
| **ICQ** | 7.4±0.0 | 35.6±2.7 | 73.6±5.0 | 110.6±3.3 |
| **TD3+BC** | 7.4±0.0 | 27.1±5.5 | 75.5±3.7 | 114.4±3.8 |
| **ICQL** | 7.4±0.0 | 41.2±10.1 | 50.4±10.8 | 64.2±24.9 |
| **OMAR** | 13.5±7.0 | 57.7±5.1 | 80.4±10.2 | 113.5±4.3 |
| **MACQL** | 5.3±0.5 | 37.0±7.1 | 51.5±26.7 | 50.1±20.1 |
| **IQL** | 7.4±0.0 | 58.8±6.8 | **81.3±3.7** | 115.6±4.2 |
| **AWAC** | 7.3±0.0 | 30.9±1.6 | 71.2±4.2 | 113.3±4.1 |
| **CFCQL** | **39.7±4.0** | **59.5±8.2** | 80.5±9.6 | **118.5±4.9** |

Except for the counterfactual Q function, we also analyze whether the counterfactual treatment in CFCQL can be incorporated in other components and help further improvement in Appendix D.4. We find that the counterfactual policy improvement, i.e., the policy improvement by Eq. 10 rather than using MADDPG's PI, is critical for our method.

## 5.4   StarCraft II

Table 3: Averaged test winning rate of CFCQL and baselines in StarCraft II micromanagement tasks.

| Map | Dataset | CFCQL | MACQL | MAICQ | OMAR | MADTKD | BC | IQL | AWAC |
|---|---|---|---|---|---|---|---|---|---|
| 2s3z | medium | **0.40±0.10** | 0.17±0.08 | 0.18±0.02 | 0.15±0.04 | 0.18±0.03 | 0.16±0.07 | 0.16±0.04 | 0.19±0.05 |
| | medium_replay | **0.55±0.07** | 0.12±0.08 | 0.41±0.06 | 0.24±0.09 | 0.36±0.07 | 0.33±0.04 | 0.33±0.06 | 0.39±0.05 |
| | expert | **0.99±0.01** | 0.58±0.34 | 0.93±0.04 | 0.95±0.04 | **0.99±0.02** | 0.97±0.02 | 0.98±0.03 | 0.97±0.03 |
| | mixed | 0.84±0.09 | 0.67±0.17 | **0.85±0.07** | 0.60±0.04 | 0.47±0.08 | 0.44±0.06 | 0.19±0.04 | 0.14±0.04 |
| 3s_vs_5z | medium | **0.28±0.03** | 0.09±0.06 | 0.03±0.01 | 0.00±0.00 | 0.01±0.01 | 0.08±0.02 | 0.20±0.05 | 0.19±0.03 |
| | medium_replay | **0.12±0.04** | 0.01±0.01 | 0.01±0.02 | 0.00±0.00 | 0.01±0.01 | 0.01±0.01 | 0.04±0.04 | 0.08±0.05 |
| | expert | **0.99±0.01** | 0.92±0.05 | 0.91±0.04 | 0.64±0.08 | 0.67±0.08 | 0.98±0.02 | **0.99±0.01** | **0.99±0.02** |
| | mixed | **0.60±0.14** | 0.17±0.10 | 0.10±0.04 | 0.00±0.00 | 0.14±0.08 | 0.21±0.04 | 0.20±0.06 | 0.18±0.03 |
| 5m_vs_6m | medium | **0.29±0.05** | 0.01±0.01 | 0.26±0.03 | 0.19±0.06 | 0.21±0.04 | 0.28±0.37 | 0.25±0.02 | 0.22±0.04 |
| | medium_replay | **0.22±0.06** | 0.16±0.08 | 0.18±0.04 | 0.03±0.02 | 0.16±0.04 | 0.18±0.06 | 0.18±0.04 | 0.18±0.04 |
| | expert | **0.84±0.03** | 0.01±0.01 | 0.72±0.05 | 0.33±0.06 | 0.58±0.07 | 0.82±0.04 | 0.77±0.03 | 0.75±0.02 |
| | mixed | 0.76±0.07 | 0.01±0.01 | 0.67±0.08 | 0.10±0.10 | 0.21±0.05 | 0.21±0.12 | 0.76±0.06 | **0.78±0.02** |
| 6h_vs_8z | medium | 0.41±0.04 | 0.01±0.01 | 0.19±0.04 | 0.04±0.03 | 0.22±0.07 | 0.40±0.03 | 0.40±0.05 | **0.43±0.06** |
| | medium_replay | **0.21±0.05** | 0.08±0.04 | 0.07±0.04 | 0.00±0.00 | 0.12±0.05 | 0.11±0.04 | 0.17±0.03 | 0.14±0.04 |
| | expert | **0.7±0.06** | 0.00±0.00 | 0.24±0.08 | 0.01±0.01 | 0.48±0.08 | 0.60±0.04 | 0.67±0.03 | 0.67±0.03 |
| | mixed | **0.49±0.08** | 0.01±0.01 | 0.05±0.03 | 0.00±0.00 | 0.25±0.07 | 0.27±0.06 | 0.36±0.05 | 0.35±0.06 |

We further validate CFCQL's universality through complex experiments on the StarCraft II Micromanagement Benchmark [40], encompassing four maps with varying agent counts and difficulties: 2s3z, 3s_vs_5z, 5m_vs_6m, and 6h_vs_8z. As no pre-existing datasets exist for these tasks, we generate them ourselves, with dataset creation details provided in the Appendix C.2.

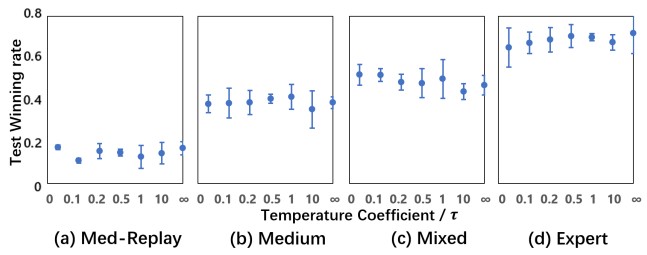

Figure 3: Hyperparameters examination on the temperature coefficient in different types of datasets.

Table 3 presents the average test winning rates of various algorithms on different datasets. MACQL's performance depends on agent count and environmental difficulty, only succeeding in 2s3z and 3s_vs_5z. MAICQ and OMAR perform well across many datasets but can not tackle all tasks and significantly

struggle in 6h_vs_8z, a highly challenging map. MADTKD, employing supervised learning and knowledge distillation, works well but seldom surpasses BC. IQL and AWAC are competitive baselines but they still fall short compared to CFCQL in most datasets. CFCQL significantly outperforms all baselines on most datasets, achieving state-of-the-art results, with its success attributed to moderate and appropriate conservatism compared to MACQL and other baselines.

**Temperature Coefficient.** To study the effect of temperature coefficient $\tau$, the key hyperparameter in computing the $\lambda_i$, Fig.3(a) shows the testing winning rate of CFCQL with different $\tau$s on each kind of dataset of map 6h_vs_8z. As shown, CFCQL is not sensitive to this hyperparameter and we find that the best value of $\tau$ is usually greater than 0 while lower than infinity, showing that moderate imbalance can be more effective as expected in previous section.

## 6 Discussion and Conclusion

### 6.1 Broader Impacts

Our proposed method holds potential for application in real-world multi-agent systems, such as intelligent warehouse management or medical treatment. However, directly implementing the derived policy might entail risks due to the domain gap between the training virtual datasets and real-world scenarios. To mitigate potential hazards, it is crucial for practitioners to operate the policy under human supervision, ensuring that undesirable outcomes are avoided by limiting the available options.

### 6.2 Limitations

Here we discuss some limitations about CFCQL. In the case of discrete action space, since CFCQL uses QMIX as the backbone, it inherits the Individual-global-max principle [42], which means it cannot solve tasks that are not factorizable. On continuous action space, the counterfactual policy update used in CFCQL allows for updating only one agent's policy for each sample, which may lead to lower convergence speed compared to methods with independent learning.

### 6.3 Conclusion

In this paper, we study the offline MARL problem which is practical and challenging but lack of enough attention. We demonstrate from theories and experiments that naively extending conservative offline RL algorithm to multi-agent setting leads to over-pessimism as the exponentially explosion of action space with the increasing number of agents which hurts the performance. To address this issue, we propose a counterfactual way to make conservative value estimation while maintaining the CTDE paradigm. It is theoretically proved that the proposed CFCQL enjoys performance guarantees independent of number of agents, which can avoid overpessimism caused by the exponentially growth of joint policy space. The results of experiments with discrete or continuous action space show that our method achieve superior performance to the state-of-the-art. Some ablation study is also made to propose further understanding of CFCQL. The idea of counterfactual treatment may also be incorporated with other offline RL and multi-agent RL algorithms, which deserves further investigation both theoretically and empirically.

## 7 Acknowledgement

This work was supported by the National Key R&D Program of China under Grant 2018AAA0102801.

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

# A Detailed Proof

## A.1 Proof of Theorem 4.1

*Proof.* Similar to the proof of Theorem 3.2 in Kumar et al. [21], we first prove this theorem in the absence of sampling error, and then incorporate sampling error at the end. By set the derivation of the objective in Eq. 4 to zero, we can compute the Q-function update induced in the exact, tabular setting($\mathcal{T}^{\boldsymbol{\pi}} = \hat{\mathcal{T}}^{\boldsymbol{\pi}}$ and $\boldsymbol{\pi}_\beta(\mathbf{a}|s) = \hat{\boldsymbol{\pi}}_\beta(\mathbf{a}|s)$).

$$\forall s, \boldsymbol{a}, k, \ \hat{Q}^{k+1}(s, \boldsymbol{a}) = \mathcal{T}^{\boldsymbol{\pi}} \hat{Q}^k(s, \boldsymbol{a}) - \alpha \left[ \Sigma_{i=1}^n \lambda_i \frac{\mu^i}{\pi_\beta^i} - 1 \right] \tag{A.1}$$

Then, the value of the policy, $\hat{V}^{k+1}$ can be proved to be underestimated, since:

$$\hat{V}^{k+1}(s) = \mathbb{E}_{\boldsymbol{a} \sim \boldsymbol{\pi}(\boldsymbol{a}|s)} \left[ \hat{Q}^{\boldsymbol{\pi}}(s, \boldsymbol{a}) \right] = \mathcal{T}^{\boldsymbol{\pi}} \hat{V}^k(s) - \alpha \mathbb{E}_{\boldsymbol{a} \sim \boldsymbol{\pi}(\boldsymbol{a}|s)} \left[ \Sigma_{i=1}^n \lambda_i \frac{\mu^i}{\pi_\beta^i} - 1 \right] \tag{A.2}$$

Next, we will show that $D_{CQL}^{CF}(s) = \Sigma_a \boldsymbol{\pi}(\boldsymbol{a}|s) \left[ \Sigma_{i=1}^n \lambda_i \frac{\mu^i(a^i|s)}{\hat{\pi}_\beta^i(a^i|s)} - 1 \right]$ is always positive, when $\mu^i(a^i|s) = \pi^i(a^i|s)$:

$$D_{CQL}^{CF}(s) = \Sigma_a \boldsymbol{\pi}(\boldsymbol{a}|s) \left[ \Sigma_{i=1}^n \lambda_i \frac{\mu^i(a^i|s)}{\pi_\beta^i(a^i|s)} - 1 \right] \tag{A.3}$$

$$= \Sigma_{i=1}^n \lambda_i \left[ \Sigma_{a^i} \pi^i(a^i|s) \left[ \frac{\mu^i(a^i|s)}{\pi_\beta^i(a^i|s)} - 1 \right] \right] \tag{A.4}$$

$$= \Sigma_{i=1}^n \lambda_i \left[ \Sigma_{a^i} (\pi^i(a^i|s) - \pi_\beta^i(a^i|s) + \pi_\beta^i(a^i|s)) \left[ \frac{\mu^i(a^i|s)}{\pi_\beta^i(a^i|s)} - 1 \right] \right] \tag{A.5}$$

$$= \Sigma_{i=1}^n \lambda_i \left[ \Sigma_{a^i} (\pi^i(a^i|s) - \pi_\beta^i(a^i|s)) \left[ \frac{\pi^i(a^i|s) - \pi_\beta^i(a^i|s)}{\pi_\beta^i(a^i|s)} \right] + \Sigma_{a^i} \pi_\beta^i(a^i|s)) \left[ \frac{\mu^i(a^i|s)}{\pi_\beta^i(a^i|s)} - 1 \right] \right] \tag{A.6}$$

$$= \Sigma_{i=1}^n \lambda_i \left[ \Sigma_{a^i} \left[ \frac{(\pi^i(a^i|s) - \pi_\beta^i(a^i|s))^2}{\pi_\beta^i(a^i|s)} \right] + 0 \right] \ since, \forall i, \Sigma_{a^i} \pi^i(a^i|s) = \Sigma_{a^i} \pi_\beta^i(a^i|s) = 1 \tag{A.7}$$

$$\geq 0 \tag{A.8}$$

As shown above, the $D_{CQL}^{CF}(s) \geq 0$, and $D_{CQL}^{CF}(s) = 0$, iff $\pi^i(a^i|s) = \pi_\beta^i(a^i|s)$. This implies that each value iterate incurs some underestimation, i.e. $\hat{V}^{k+1}(s) \leq \mathcal{T}^{\boldsymbol{\pi}} \hat{V}^k(s)$.

We can compute the fixed point of the recursion in Equation A.2 and get the following estimated policy value:

$$\hat{V}^{\boldsymbol{\pi}}(s) = V^{\boldsymbol{\pi}}(s) - \alpha \left[ (I - \gamma P^{\boldsymbol{\pi}})^{-1} \Sigma_a \boldsymbol{\pi}(\boldsymbol{a}|s) \left[ \Sigma_{i=1}^n \lambda_i \frac{\mu^i(a^i|s)}{\hat{\pi}_\beta^i(a^i|s)} - 1 \right] \right] (s) \tag{A.9}$$

Because the $(I - \gamma P^{\boldsymbol{\pi}})^{-1}$ is non negative and the $D_{CQL}^{CF}(s) \geq 0$, it's easily to prove that in the absence of sampling error, Theorem 4.1 gives a lower bound.

**Incorporating sampling error**. According to the conclusion in Kumar et al. [21], we can directly write down the result with sampling error as follows:

$$\hat{V}^{\boldsymbol{\pi}}(s) \leq V^{\boldsymbol{\pi}}(s) - \alpha \left[ (I - \gamma P^{\boldsymbol{\pi}})^{-1} \Sigma_a \boldsymbol{\pi}(\boldsymbol{a}|s) \left[ \Sigma_{i=1}^n \lambda_i \frac{\mu^i(a^i|s)}{\hat{\pi}_\beta^i(a^i|s)} - 1 \right] \right] (s) + \left[ (I - \gamma P^{\boldsymbol{\pi}})^{-1} \frac{C_{r,T,\sigma} R_{max}}{(1 - \gamma)\sqrt{|D|}} \right] \tag{A.10}$$

So, the statement of Theorem 4.1 with sampling error is proved. Please refer to the Sec.D.3 in Kumar et al. [21] For detailed proof. Besides, the choice of $\alpha$ in this case to prevent overestimation is given by:

$$\alpha \geq \max_{s,\boldsymbol{a} \in D} \frac{C_{r,T,\sigma} R_{max}}{(1-\gamma)\sqrt{|D|}} \cdot \max_{s \in D} \left[ \Sigma_a \boldsymbol{\pi}(\boldsymbol{a}|s) \left[ \Sigma_{i=1}^{n} \lambda_i \frac{\mu^i(a^i|s)}{\hat{\pi}_{\beta}^i(a^i|s)} - 1 \right] \right]^{-1} \tag{A.11}$$

$\square$

## A.2  Proof of Theorem 4.2

*Proof.* According to the definition, we can get the formulation of $D_{CQL}^{CF}(\boldsymbol{\pi},\boldsymbol{\beta})(s)$ and $D_{CQL}(\boldsymbol{\pi},\boldsymbol{\beta})(s)$ as follow:

$$D_{CQL}^{CF}(\boldsymbol{\pi},\boldsymbol{\beta})(s) = \mathbb{E}_{\boldsymbol{a}\sim\boldsymbol{\pi}(\cdot|s)} \left( \left[ \sum_{i=1}^{n} \lambda_i \frac{\pi^i(a^i|s)}{\beta^i(a^i|s)} \right] - 1 \right) \tag{A.12}$$

$$= \sum_{i=1}^{n} \lambda_i \left( \sum_{a^i} \frac{\pi^i(a^i|s) * \pi^i(a^i|s)}{\beta^i(a^i|s)} \right) - 1 \geq 0 \tag{A.13}$$

$$D_{CQL}(\boldsymbol{\pi},\boldsymbol{\beta})(s) = \mathbb{E}_{\boldsymbol{a}\sim\boldsymbol{\pi}(\cdot|s)} \left( \left[ \frac{\boldsymbol{\pi}(\boldsymbol{a}|s)}{\boldsymbol{\beta}(\boldsymbol{a}|s)} \right] - 1 \right) \tag{A.14}$$

$$= \prod_{i=1}^{n} \left( \sum_{a^i} \frac{\pi^i(a^i|s) * \pi^i(a^i|s)}{\beta^i(a^i|s)} \right) - 1 \geq 0 \tag{A.15}$$

Then, by taking the logarithm of $D_{CQL}(\boldsymbol{\pi},\boldsymbol{\beta})(s)$, we get:

$$\ln(D_{CQL}(\boldsymbol{\pi},\boldsymbol{\beta})(s) + 1) = \sum_{i=1}^{n} \ln \left( \mathbb{E}_{a^i\sim\pi^i(\cdot|s)} \frac{\pi^i(a^i|s)}{\beta^i(a^i|s)} \right) \tag{A.16}$$

As $\Sigma_i \lambda_i = 1$, it's obvious that

$$\ln(D_{CQL}^{CF}(\boldsymbol{\pi},\boldsymbol{\beta})(s) + 1) \leq \ln \left( \sum_{a^j} \frac{\pi^j(a^j|s) * \pi^j(a^j|s)}{\beta^j(a^j|s)} \right), where \ j = \arg\max_k \mathbb{E}_{\pi^k} \frac{\pi^k}{\beta^k} \tag{A.17}$$

By combining equation A.16 and inequation A.17, we get

$$\frac{D_{CQL}(\boldsymbol{\pi},\boldsymbol{\beta})(s) + 1}{D_{CQL}^{CF}(\boldsymbol{\pi},\boldsymbol{\beta})(s) + 1} \geq \exp \left( \sum_{i=1,i\neq j}^{n} \ln \left( \mathbb{E}_{a^i\sim\pi^i(\cdot|s)} \frac{\pi^i(a^i|s)}{\beta^i(a^i|s)} \right) \right) \tag{A.18}$$

$$\geq \exp \left( \sum_{i=1,i\neq j}^{n} KL(\pi^i(s)||\beta^i(s)) \right), where \ j = \arg\max_k \mathbb{E}_{\pi^k} \frac{\pi^k}{\beta^k} \tag{A.19}$$

the second inequality is derived from the Jensen's inequality. As the Kullback-Leibler Divergence is non-negative, it's obvious that $D_{CQL}(\boldsymbol{\pi},\boldsymbol{\beta})(s) \geq D_{CQL}^{CF}(\boldsymbol{\pi},\boldsymbol{\beta})(s)$, then we can simplify the left-hand side of this inequality:

$$\frac{D_{CQL}(\boldsymbol{\pi},\boldsymbol{\beta})(s)}{D_{CQL}^{CF}(\boldsymbol{\pi},\boldsymbol{\beta})(s)} \geq \exp \left( \sum_{i=1,i\neq j}^{n} KL(\pi^i(s)||\beta^i(s)) \right), where \ j = \arg\max_k \mathbb{E}_{\pi^k} \frac{\pi^k}{\beta^k} \tag{A.20}$$

$\square$

### A.3 Proof of Equation 6

*Proof.* Similar to the proof of Lemma D.3.1 in CQL [21], $Q$ is obtained by solving a recursive Bellman fixed point equation in the empirical MDP $\hat{M}$, with an altered reward, $r(s,a) - \alpha \left[ \sum_i \lambda_i \frac{\pi^i(a^i|s)}{\beta^i(a^i|s)} - 1 \right]$, hence the optimal policy $\boldsymbol{\pi}^*(\boldsymbol{a}|s)$ obtained by optimizing the value under the CFCQL Q-function equivalently is characterized via Eq. 6. $\qquad\square$

### A.4 Proof of Theorem 4.3

*Proof.* Similar to Eq. 6, $\boldsymbol{\pi}^*_{MA}$ is equivalently obtained by solving:

$$\boldsymbol{\pi}^*_{MA}(\boldsymbol{a}|s) \leftarrow \arg\max_{\boldsymbol{\pi}} J(\boldsymbol{\pi}, \hat{M}) - \alpha \frac{1}{1-\gamma} \mathbb{E}_{s \sim d^{\boldsymbol{\pi}}_{\hat{M}}(s)}[D_{CQL}(\boldsymbol{\pi}, \boldsymbol{\beta})(s)]. \qquad (A.21)$$

Recall that $\forall s, \boldsymbol{\pi}, \boldsymbol{\beta}, D_{CQL}(\boldsymbol{\pi}, \boldsymbol{\beta})(s) \geq 0$. We have

$$\begin{aligned} J(\boldsymbol{\pi}^*_{MA}, \hat{M}) \geq & J(\boldsymbol{\pi}^*_{MA}, \hat{M}) - \alpha \frac{1}{1-\gamma} \mathbb{E}_{s \sim d^{\boldsymbol{\pi}^*_{MA}}_{\hat{M}}(s)}[D_{CQL}(\boldsymbol{\pi}^*_{MA}, \boldsymbol{\beta})(s)] \\ \geq & J(\boldsymbol{\pi}^*, \hat{M}) - \alpha \frac{1}{1-\gamma} \mathbb{E}_{s \sim d^{\boldsymbol{\pi}^*}_{\hat{M}}(s)}[D_{CQL}(\boldsymbol{\pi}^*, \boldsymbol{\beta})(s)]. \end{aligned} \qquad (A.22)$$

Then we give an upper bound of $\mathbb{E}_{s \sim d^{\boldsymbol{\pi}^*}_{\hat{M}}(s)}[D_{CQL}(\boldsymbol{\pi}^*, \boldsymbol{\beta})(s)]$. Due to the assumption that $\beta^i$ is greater than $\epsilon$ anywhere, we have

$$\begin{aligned} D_{CQL}(\boldsymbol{\pi}, \boldsymbol{\beta})(s) = & \sum_{\boldsymbol{a}} \boldsymbol{\pi}(\boldsymbol{a}|s)[\frac{\boldsymbol{\pi}(\boldsymbol{a}|s)}{\boldsymbol{\beta}(\boldsymbol{a}|s)} - 1] = \sum_{\boldsymbol{a}} \boldsymbol{\pi}(\boldsymbol{a}|s)[\frac{\boldsymbol{\pi}(\boldsymbol{a}|s)}{\prod_{i=1}^n \beta^i(a^i|s)} - 1] \\ \leq & \left( \frac{1}{\epsilon^n} \sum_{\boldsymbol{a}} \boldsymbol{\pi}(\boldsymbol{a}|s)[\boldsymbol{\pi}(\boldsymbol{a}|s)] \right) - 1 \leq \frac{1}{\epsilon^n} - 1. \end{aligned} \qquad (A.23)$$

Combining Eq. A.22 and Eq. A.23, we can get

$$J(\boldsymbol{\pi}^*_{MA}, \hat{M}) \geq J(\boldsymbol{\pi}^*, \hat{M}) - \frac{\alpha}{1-\gamma}(\frac{1}{\epsilon^n} - 1) \qquad (A.24)$$

Recall the sampling error proved in [21] and referred to above in (A.10), we can use it to bound the performance difference for any $\boldsymbol{\pi}$ on true and empirical MDP by

$$|J(\boldsymbol{\pi}, M) - J(\boldsymbol{\pi}, \hat{M})| \leq \frac{C_{r,T,\delta} R_{max}}{(1-\gamma)^2} \sum_s \frac{\rho(s)}{\sqrt{|D(s)|}}, \qquad (A.25)$$

then let $sampling\ error := 2 \cdot \frac{C_{r,T,\sigma} R_{max}}{(1-\gamma)^2} \sum_s \frac{\rho(s)}{\sqrt{|D(s)|}}$, and incorporate it into (A.24), we get

$$J(\boldsymbol{\pi}^*_{MA}, M) \geq J(\boldsymbol{\pi}^*, M) - \frac{\alpha}{1-\gamma}(\frac{1}{\epsilon^n} - 1) - sampling\ error \qquad (A.26)$$

where $sampling\ error$ is a constant dependent on the MDP itself and D. Note that during the proof we do not take advantage of the nature of $\boldsymbol{\pi}^*$. Actually $\boldsymbol{\pi}^*$ can be replaced by any policy $\boldsymbol{\pi}$. The reason we use $\boldsymbol{\pi}^*$ is that it can give that largest lower bound, resulting in the best policy improvement guarantee. Similarly, $D^{CF}_{CQL}$ can be bounded by $\frac{1}{\epsilon} - 1$:

$$\begin{aligned} D^{CF}_{CQL}(\boldsymbol{\pi}, \boldsymbol{\beta})(s) = & \sum_{i=1}^n \lambda_i \sum_{a^i} \pi^i(a^i|s)[\frac{\pi^i(a^i|s)}{\beta^i(a^i|s)} - 1] \\ \leq & \left( \frac{1}{\epsilon} \sum_{i=1}^n \lambda_i \sum_{a^i} \pi^i(a^i|s)[\pi^i(a^i|s)] \right) - 1 \\ \leq & \frac{1}{\epsilon} \left( \sum_{i=1}^n \lambda_i \right) - 1 = \frac{1}{\epsilon} - 1. \end{aligned} \qquad (A.27)$$

$\qquad\square$

## A.5 Proof of Theorem 4.4

We first show the theorem of safe policy improvement guarantee for MACQL and CFCQL, separately. Then we compare these two gaps.

MACQL has a safe policy improvement guarantee related to the number of agents $n$:

**Theorem A.1.** *Given the discounted marginal state-distribution* $d_{\hat{M}}^{\pi}$, *we define* $\mathcal{B}(\pi, D) = \mathbb{E}_{s \sim d_{\hat{M}}^{\pi}}[\sqrt{D(\pi, \beta)(s) + 1}]$. *The policy* $\pi_{MA}^*(a|s)$ *is a* $\zeta^{MA}$-*safe policy improvement over* $\beta$ *in the actual MDP* $M$, *i.e.,* $J(\pi_{MA}^*, M) \geq J(\beta, M) - \zeta^{MA}$, *where* $\zeta^{MA} = 2\left(\frac{C_{r,\delta}}{1-\gamma} + \frac{\gamma R_{\max} C_{T,\delta}}{(1-\gamma)^2}\right) \cdot \frac{\sqrt{|A|}}{\sqrt{|\mathcal{D}(s)|}} \mathcal{B}(\pi_{MA}^*, D_{CQL}) + \frac{\alpha}{1-\gamma}(\frac{1}{\epsilon^n} - 1) - (J(\pi^*, \hat{M}) - J(\hat{\beta}, \hat{M}))$.

*Proof.* We can first get a $J(\pi_{MA}^*, \hat{M})$-related policy improvement guarantee following the proof of Theorem 3.6 in Kumar et al. [21]:

$$
\begin{aligned}
J(\pi_{MA}^*, M) \geq & J(\beta, M) - \left(2\left(\frac{C_{r,\delta}}{1-\gamma} + \frac{\gamma R_{\max} C_{T,\delta}}{(1-\gamma)^2}\right) \cdot \frac{\sqrt{|A|}}{\sqrt{|\mathcal{D}(s)|}} \mathcal{B}(\pi_{MA}^*, D_{CQL}) \right. \\
& \left. - (J(\pi_{MA}^*, \hat{M}) - J(\hat{\beta}, \hat{M}))\right)
\end{aligned}
\tag{A.28}
$$

According to Eq. A.21, $\pi_{MA}^*$ is obtained by optimizing $J(\pi, \hat{M})$ with a $D_{CQL}$-related regularizer. And Theorem 4.3 shows that $D_{CQL}$ can be extremely large when the team size expands, which may severely change the optimization objective and affects the shape of the optimization plane. Therefore, $J(\pi_{MA}^*, \hat{M})$ may be extremely low, and keeping $J(\pi_{MA}^*, \hat{M})$ in Eq. A.28 results in a mediocre policy improvement guarantee. To bound $J(\pi_{MA}^*, \hat{M})$, we introduce Eq. A.24 into Eq. A.28, we get the following:

$$
\begin{aligned}
J(\pi_{MA}^*, M) \geq & J(\beta, M) - \left(2\left(\frac{C_{r,\delta}}{1-\gamma} + \frac{\gamma R_{\max} C_{T,\delta}}{(1-\gamma)^2}\right) \cdot \frac{\sqrt{|A|}}{\sqrt{|\mathcal{D}(s)|}} \mathcal{B}(\pi_{MA}^*, D_{CQL}) \right. \\
& \left. + \frac{\alpha}{1-\gamma}(\frac{1}{\epsilon^n} - 1) - (J(\pi^*, \hat{M}) - J(\hat{\beta}, \hat{M}))\right)
\end{aligned}
\tag{A.29}
$$

This complete the proof. $\qquad\square$

We can get a similar $\zeta^{CF}$ satisfying $J(\pi_{CF}^*, M) \geq J(\beta, M) - \zeta^{CF}$ for CFCQL, which is independent of $n$:

$$
\zeta^{CF} = 2\left(\frac{C_{r,\delta}}{1-\gamma} + \frac{\gamma R_{\max} C_{T,\delta}}{(1-\gamma)^2}\right) \cdot \frac{\sqrt{|A|}}{\sqrt{|\mathcal{D}(s)|}} \mathcal{B}(\pi_{CF}^*, D_{CQL}^{CF}) + \frac{\alpha}{1-\gamma}(\frac{1}{\epsilon} - 1) - (J(\pi^*, \hat{M}) - J(\hat{\beta}, \hat{M}))
\tag{A.30}
$$

Then we can prove Theorem 4.4.

*Proof.* Subtract $\zeta^{CF}$ from $\zeta^{MA}$, and we get:

$$
\zeta^{MA} - \zeta^{CF} = 2\left(\frac{C_{r,\delta}}{1-\gamma} + \frac{\gamma R_{\max} C_{T,\delta}}{(1-\gamma)^2}\right) \frac{\sqrt{|A|}}{\sqrt{|\mathcal{D}(s)|}} \left(\mathcal{B}(\pi_{MA}^*, D_{CQL}) - \mathcal{B}(\pi_{CF}^*, D_{CQL}^{CF})\right) + \frac{\alpha}{1-\gamma}(\frac{1}{\epsilon^n} - \frac{1}{\epsilon})
\tag{A.31}
$$

Let the right side $\geq 0$, and we can get

$$
n \geq \log_{\frac{1}{\epsilon}}\left[\max\left(1, \frac{1}{\epsilon} + \frac{2}{\alpha} \frac{\sqrt{|A|}}{\sqrt{|\mathcal{D}(s)|}}\left(C_{r,\delta} + \frac{\gamma R_{\max} C_{T,\delta}}{1-\gamma}\right) \cdot \left[\mathcal{B}(\pi_{CF}^*, D_{CQL}^{CF}) - \mathcal{B}(\pi_{MA}^*, D_{CQL})\right]\right)\right]
\tag{A.32}
$$

According to Theorem 4.3,

$$
\mathcal{B}(\pi_{CF}^*, D_{CQL}^{CF}) = \mathbb{E}_{s \sim d_{\hat{M}}^{\pi_{CF}^*}}[\sqrt{D_{CQL}^{CF}(\pi_{CF}^*, \beta)(s) + 1}] \leq \mathbb{E}_{s \sim d_{\hat{M}}^{\pi_{CF}^*}}[\sqrt{\frac{1}{\epsilon} - 1 + 1}] = \frac{1}{\sqrt{\epsilon}}
\tag{A.33}
$$

In the meantime, we have

$$\mathcal{B}\left(\boldsymbol{\pi}^*_{CF}, D^{CF}_{CQL}\right) = \mathbb{E}_{s\sim d^{\boldsymbol{\pi}^*_{MA}}_{\hat{M}}}[\sqrt{D_{CQL}(\boldsymbol{\pi}^*_{MA}, \boldsymbol{\beta})(s) + 1}] \geq \mathbb{E}_{s\sim d^{\boldsymbol{\pi}^*_{MA}}_{\hat{M}}}[\sqrt{D_{CQL}(\boldsymbol{\beta}, \boldsymbol{\beta})(s) + 1}] = 1 \tag{A.34}$$

Therefore, we can relax the lower bound of $n$ to a constant that

$$n \geq \log_{\frac{1}{\epsilon}}\left(\frac{1}{\epsilon} + \frac{2}{\alpha}\frac{\sqrt{|A|}}{\sqrt{|\mathcal{D}(s)|}}(C_{r,\delta} + \frac{\gamma R_{\max}C_{T,\delta}}{1-\gamma}) \cdot (\frac{1}{\sqrt{\epsilon}} - 1)\right) \tag{A.35}$$

$\square$

# B  Implement Details

## B.1  Derivation of the Update Rule

To utilize the Eq. 4 for policy optimization, following the analysis in the Section 3.2 in Kumar et al. [21], we formally define optimization problems over each $\mu^i(a^i|s)$ by adding a regularizer $R(\mu^i)$. As shown below, we mark the modifications from the Eq. 4 in red.

$$\min_Q \max_{\boldsymbol{\mu}} \alpha\left[\sum_{i=1}^n \lambda_i \mathbb{E}_{s\sim\mathcal{D}, a^i\sim\mu^i, \boldsymbol{a}^{-i}\sim\boldsymbol{\beta}^{-i}}[Q(s,\boldsymbol{a})] - \mathbb{E}_{s\sim\mathcal{D}, \boldsymbol{a}\sim\boldsymbol{\beta}}[Q(s,\boldsymbol{a})]\right]$$
$$+ \frac{1}{2}\mathbb{E}_{s,\boldsymbol{a},s'\sim\mathcal{D}}\left[(Q(s,\boldsymbol{a}) - \hat{\mathcal{T}}^{\boldsymbol{\pi}}\hat{Q}_k(s,\boldsymbol{a}))^2\right] + \sum_{i=1}^n \lambda_i R(\mu^i), \tag{B.36}$$

By choosing different regularizer, there are a variety of instances within CQL family. As recommended in Kumar et al. [21], we choose $R(\mu^i)$ to be the KL-divergence against a Uniform distribution over action space, i.e., $R(\mu^i) = -D_{KL}(\mu^i, Unif(a^i))$. Then we can get the following objective for $\mu^i$:

$$\max_{\mu^i} \mathbb{E}_{x\sim\mu^i(x)}[f(x)] + \mathcal{H}(\mu^i), \quad s.t. \sum_x \mu^i(x) = 1, \mu^i(x) \geq 0, \forall x, \tag{B.37}$$

where $\forall s, f(x) = Q(s,x,\boldsymbol{a}^{-i})$. The optimal solution is:

$$\mu^{i*}(x) = \frac{1}{Z}\exp(f(x)), \tag{B.38}$$

where $Z$ is the normalization factor, i.e., $Z = \sum_x \exp(f(x))$. Plugging this back into Eq. B.36, we get:

$$\min_Q \alpha\mathbb{E}_{s\sim\mathcal{D}}\left[\sum_{i=1}^n \lambda_i \mathbb{E}_{\boldsymbol{a}^{-i}\sim\boldsymbol{\beta}^{-i}}[\log\sum_{a^i}\exp(Q(s,\boldsymbol{a}))] - \mathbb{E}_{\boldsymbol{a}\sim\boldsymbol{\beta}}[Q(s,\boldsymbol{a})]\right]$$
$$+ \frac{1}{2}\mathbb{E}_{s,\boldsymbol{a},s'\sim\mathcal{D}}\left[(Q(s,\boldsymbol{a}) - \hat{\mathcal{T}}^{\boldsymbol{\pi}_k}\hat{Q}_k(s,\boldsymbol{a}))^2\right]. \tag{B.39}$$

## B.2  Details for Computing $\lambda$

To compute $\lambda$, we need an explicit expression of $\pi^i$ and $\beta^i$. In the setting of discrete action space, as we use Q-learning, $\pi^i$ can be expressed by the Boltzman policy, i.e.

$$\pi^i(a^i_j) = \frac{\exp\left(\mathbb{E}_{\boldsymbol{a}^{-i}\sim\boldsymbol{\beta}^{-i}}Q(s,a^i_j,\boldsymbol{a}^{-i})\right)}{\sum_k \exp\left(\mathbb{E}_{\boldsymbol{a}^{-i}\sim\boldsymbol{\beta}^{-i}}Q(s,a^i_k,\boldsymbol{a}^{-i})\right)} \tag{B.40}$$

We use behaviour cloning to pre-train a parameterized $\boldsymbol{\beta}(s)$ with a three-level fully-connected network and MLE(Maximum Likelihood Estimation) loss.

With the explicit expression of $\pi^i$ and $\beta^i$, we can directly compute $\lambda$ with Eq. 8 and Eq. 9. While, in practice, we find the $\mathbb{E}_{\pi^i}\frac{\pi^i(s)}{\beta^i(s)}$ may introduce extreme variance as its large scale and fluctuations,

which will hurt the performance. Instead, we take the logarithm of it and further reduced it to the Kullback-Leibler Divergence as follow:

$$\forall i, s, \lambda_i(s) = \frac{\exp\left(-\tau D_{KL}(\pi^i(s)||\beta^i(s))\right)}{\sum_{j=1}^{n} \exp\left(-\tau D_{KL}(\pi^j(s)||\beta^j(s))\right)}, \tag{B.41}$$

For continuous action space, we use the deterministic policy like in MADDPG, whose policy distribution can be regared as a Dirac delta function. Therefore, we approximate $\mathbb{E}_{\pi^j} \frac{\pi^j(s)}{\beta^j(s)}$ by the following:

$$\mathbb{E}_{\pi^j} \frac{\pi^j(s)}{\beta^j(s)} \approx \frac{1}{\beta^j(\pi^j(s)|s)} \tag{B.42}$$

Then we need to obtain an explicit expression of $\beta^i$. We first train a VAE [17] from the dataset to obtain the lower bound of $\beta^i$. Let $p_\phi(a, z|s)$ and $q_\varphi(z|a, s)$ be the decoder and the encoder of the trained VAE, respectively. According to Wu et al. [48], $\beta^j(a^j|s)$ can be explicitly estimated by (We omit the superscript $j$ for brevity):

$$\log \beta_\phi(a \mid s) = \log \mathbb{E}_{q_\varphi(z|a,s)} \left[ \frac{p_\phi(a, z \mid s)}{q_\varphi(z \mid a, s)} \right]$$

$$\approx \mathbb{E}_{z^{(l)} q_\varphi(z|a,s)} \left[ \log \frac{1}{L} \sum_{l=1}^{L} \frac{p_\phi\left(a, z^{(l)} \mid s\right)}{q_\varphi\left(z^{(l)} \mid a, s\right)} \right] \tag{B.43}$$

$$\overset{\text{def}}{=} \widehat{\log \pi_\beta}(a \mid s; \varphi, \phi, L).$$

Therefore, we can sample from the VAE $L$ times to estimate $\beta^i$. The sampling error reduces as $L$ increases.

## C  Experimental Details

### C.1  Tasks

$Equal\_Line$ is a multi-agent task which we design by simplify the space shape of $Equal\_Space$ to one-dimension. There are $n$ agents and they are randomly initialized to the interval $[0, 2]$. The state space is a a one-dimensional bounded region in $[0, \max(10, 2*n)]$ and the local action space is a discrete, eleven-dimensional space, i.e. $[0, -0.01, -0.05, -0.1, -0.5, -1, 0.01, 0.05, 0.1, 0.5, 1]$, which represents the moving direction and distance at each step. The reward is shared by the agents and formulated as $10 * (n - 1) \frac{min\_dis - last\_step\_min\_dis}{line\_length}$, which will spur the agents to cooperate to spread out and keep the same distance between each other.

For Multi-agent Particle Environment and Multi-agent Mujoco, we adopt the open-source implementations from Lowe et al. [27][3] and Peng et al. [35][4] respectively. And we use the datasets and the adversary agents provided by Pan et al. [34].

For StarCraft II Micromanagement Benchmark, we use the open-source implementation from Samvelyan et al. [40][5] and choose four maps with different difficulty and number of agents as the experimental scenarios, which is summarized in Table 4. We construct our own datasets with QMIX [37] by collecting training or evaluating data.

### C.2  StarCraft II datasets collection

The datasets are made based on the training process or trained model of QMIX[37]. Specially, the $Medium$ or $Expert$ datasets are sampled by executing a partially-pretrained policy with a medium performance level or a fully-pretrained policy. The $Medium - Replay$ datasets are exactly the replay buffer during training until the policy reaches the medium performance. The $Mixed$ datasets are the equal mixture of $Medium$ and $Expert$ datasets. All datasets contain five thousand trajectories, except for the $Medium - Replay$.

---

[3] https://github.com/openai/multiagent-particle-envs
[4] https://github.com/schroederdewitt/multiagent_mujoco
[5] https://github.com/oxwhirl/smac

Table 4: The details of tested maps in the StarCraft II micromanagement benchmark

| Maps | Agents | Enemies | Difficulty |
|------|--------|---------|------------|
| 2s3z | 2 Stalkers & 3 Zealots | 2 Stalkers & 3 Zealots | Easy |
| 3s_vs_5z | 3 Stalkers | 5 Zealots | Easy |
| 5m_vs_6m | 5 Marines | 6 Marines | Hard |
| 6h_vs_8z | 6 Hydralisks | 8 Zealots | Super Hard |

## C.3 Baselines

**BC**: behavior cloning. In discrete action space, we train a three-level MLP network with MLE loss. In continuous action space, we use the method of explicit estimation of behavior density in Wu et al. [48], which is modified from a VAE [17] estimator. **TD3-BC**[11]: One of the SOTA single agent offline algorithm, simply adding the BC term to TD3 [12]. We use the open-source implementation[6] and modify it to a CTDE version with centralised critic. **IQL**[18] and **AWAC**[31]: variants of advantage weighted behaviour cloning. We refer to the open-source implementation[7] and implement a CTDE version similar to TD3-BC. **MACQL**:naive extension of conservative Q-learning, as proposed in Sec. 3.3. We implement it based on the open-source implementation[8]. As the joint action space is enormous, we sample $N$ actions for the logsumexp operation. **MAICQ**[52]:multi-agent version of implicit constraint Q-learning by propose the decomposed multi-agent joint-policy under implicit constraint. We use the open-source implementation[9] in discrete action space and cite the experimental results in continuous action space from Pan et al. [34]. **OMAR**[34]:uses zeroth-order optimization for better coordination among agents' policies, based on independent CQL (**ICQL**). We cite the experimental results in continuous action space from Pan et al. [34] and implement a version in discrete action space based on the open-source implementation[10]. **MADTKD**[45]:uses decision transformer to represent each agent's policy and trains with knowledge distillation. As lack of open-source implementation, We implement it based on the open-source implementation[11] of another Decision Transformer based method **MADT**[28].

## C.4 Resources

We use 2 servers to run all the experiments. Each one has 8*NVIDIA RTX 3090 GPUs, and 2*AMD 7H12 CPUs. Each setting is repeated for 5 seeds. For one seed in SC2, it takes about 1.5 hours. For MPE, 10 minutes is enough. The experiments on MaMuJoCo cost the most, about 5 hours for each seed.

## C.5 Code, Hyper-parameters and Reproducibility

Please refer to this repository[12] for the code, datasets and the hyper-parameters of our method. For each dataset number $0, 1, 2, 3, 4$, we use the seed $0, 1, 2, 3, 4$, respectively.

# D More results

## D.1 Complete Results on MPE

Table 5 shows the complete results of our methods and more baselines on Multi-agent Particle Environment. Some results are cited from Pan et al. [34].

---

[6] https://github.com/sfujim/TD3_BC
[7] https://github.com/tinkoff-ai/CORL
[8] https://github.com/aviralkumar2907/CQL
[9] https://github.com/YiqinYang/ICQ
[10] https://github.com/ling-pan/OMAR
[11] https://github.com/ReinholdM/Offline-Pre-trained-Multi-Agent-Decision-Transformer
[12] https://github.com/thu-rllab/CFCQL

Table 5: Complete results on Multi-agent Particle Environment.

| Env | Dataset | MAICQ | MATD3-BC | ICQL | OMAR | MACQL | IQL | AWAC | CFCQL |
|---|---|---|---|---|---|---|---|---|---|
| CN | Random | 6.3±3.5 | 9.8±4.9 | 24.0±9.8 | 34.4±5.3 | 45.6±8.7 | 5.5±1.1 | 0.5±3.7 | **62.2±8.1** |
| | Medium-replay | 13.6±5.7 | 15.4±5.6 | 20.0±8.4 | 37.9±12.3 | 25.5±5.9 | 10.8±4.5 | 2.7±3 | **52.2±9.6** |
| | Medium | 29.3±5.5 | 29.3±4.8 | 34.1±7.2 | 47.9±18.9 | 14.3±20.2 | 28.2±3.9 | 25.7±4.1 | **65.0±10.2** |
| | Expert | 104.0±3.4 | 108.3±3.3 | 98.2±5.2 | **114.9±2.6** | 12.2±31 | 103.7±2.5 | 103.3±3.5 | 112±4 |
| PP | Random | 2.2±2.6 | 5.7±3.5 | 5.0±8.2 | 11.1±2.8 | 25.2±11.5 | 1.3±1.6 | 0.2±1.0 | **78.5±15.6** |
| | Medium-replay | 34.5±27.8 | 28.7±20.9 | 24.8±17.3 | 47.1±15.3 | 11.9±9.2 | 23.2±12 | 8.3±5.3 | **71.1±6** |
| | Medium | 63.3±20.0 | 65.1±29.5 | 61.7±23.1 | 66.7±23.2 | 55±43.2 | 53.6±19.9 | 50.9±19.0 | **68.5±21.8** |
| | Expert | 113.0±14.4 | 115.2±12.5 | 93.9±14.0 | 116.2±19.8 | 108.4±21.5 | 109.3±10.1 | 106.5±10.1 | **118.2±13.1** |
| World | Random | 1.0±3.2 | 2.8±5.5 | 0.6±2.0 | 5.9±5.2 | 11.7±11 | 2.9±4.0 | -2.4±2.0 | **68±20.8** |
| | Medium-replay | 12.0±9.1 | 17.4±8.1 | 29.6±13.8 | 42.9±19.5 | 13.2±16.2 | 41.5±9.5 | 8.9±5.1 | **73.4±23.2** |
| | Medium | 71.9±20.0 | 73.4±9.3 | 58.6±11.2 | 74.6±11.5 | 67.4±48.4 | 70.5±15.3 | 63.9±14.2 | **93.8±31.8** |
| | Expert | 109.5±22.8 | 110.3±21.3 | 71.9±28.1 | 110.4±25.7 | 99.7±31 | 107.8±17.7 | 107.6±15.6 | **119.7±26.4** |

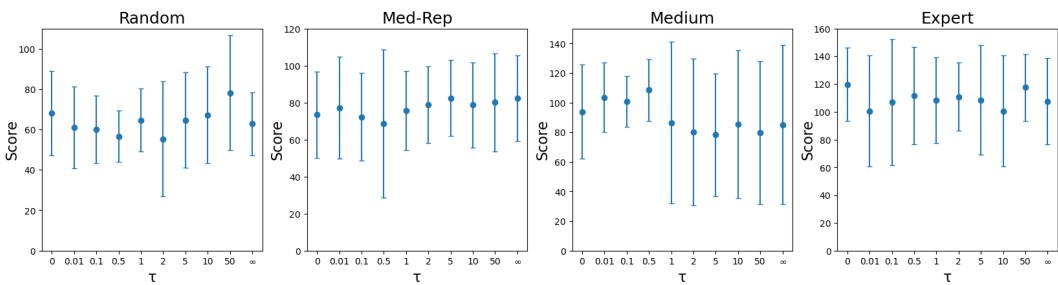

Figure 4: Ablations of $\tau$ on World.

## D.2 Temperature Coefficient in Continuous Action Space

We carry out ablations of $\tau$ on MPE's map World in Fig. 4. We find that although. the best $\tau$ differs in different datasets, the overall performance is not sensitive to $\tau$, which verifies the theoretical analysis that any simplex of $\lambda$ that $\sum_{i=1}^{n} \lambda_i = 1$ can induce an underestimated value function.

## D.3 Ablation on CQL $\alpha$

We carry out ablations of $\alpha$ on MPE's map World in Fig. 5. We find that $\alpha$ plays a more important role for team performance on narrow distributions (e.g., $Expert$ and $Medium$) than that on wide distributions (e.g., $Random$ and $Medium - Replay$).

## D.4 Component Analysis on Counterfactual style

In the environment MaMuJo, except for the counterfactual Q function, we also analyze whether the conuterfactual treatment in CFCQL can be incorporated in other components and help further improvement in Table 6. We find that the counterfactual policy improvement is critical for this

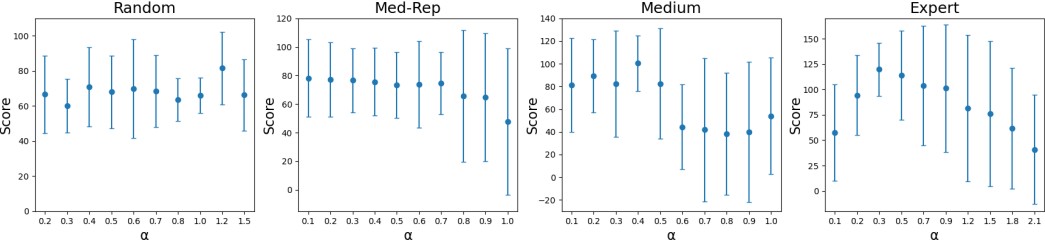

Figure 5: Ablations of $\alpha$ on World.

Table 6: Component Analysis on MaMuJoCo. CF_T: computing target Q by $\mathbb{E}_{i\sim\text{Unif}(1,n)}\mathbb{E}_{s',\boldsymbol{a}^{-i}\sim\mathcal{D},a^i\sim\pi^i}Q_{\hat{\theta}}(s,\boldsymbol{a})$. CF_P: the policy improvement (PI) by Eq. 10, otherwise using MADDPG's PI.

| Dataset | Default | +CF_T | -CF_P | MACQL |
|---|---|---|---|---|
| Random | 39.7±4.0 | **48.7±1.8** | 23.9±9.2 | 5.3±0.5 |
| Med-Rep | **59.5±8.2** | 58.9±9.6 | 43.5±5.6 | 36.7±7.1 |
| Medium | **80.5±9.6** | 76.2±12.1 | 43.8±7.8 | 51.5±26.7 |
| Expert | **118.5±4.9** | 118.1±6.9 | 3.7±3.1 | 50.1±20.1 |

environment. With CF_P, the method shows great performance gain on narrow data distribution, e.g., the $Expert$ dataset.

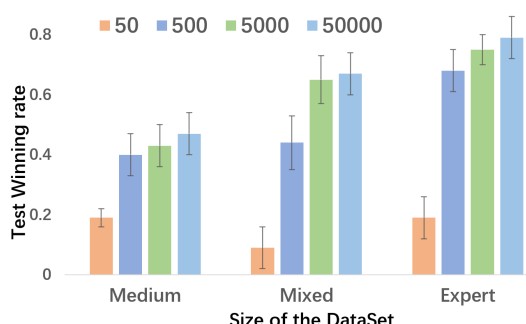

Figure 6: Hyperparameters examination on the size of datasets.

## D.5 Analysis on Size of Dataset

An additional study examines dataset size effects. We generate a full dataset with $50,000$ trajectories for each type in map $6h\_vs\_8z$, creating smaller datasets by sampling $5, 50, 500, 5000$ trajectories. Fig. 6 displays CFCQL's testing winning rate for varying dataset sizes. It's notable that to ensure fairness, the maximum number of training steps for all datasets and algorithms on the SMAC environment is fixed at 1e7. In this additional study, however, we trained the CFCQL algorithm until convergence to eliminate the impact of underfitting. The results demonstrate that larger datasets contribute to improved convergence performances, thus confirming the scalability of CFCQL for larger data samples.

# E Discussion

## E.1 Overestimation in Offline RL

We offer an intuitive explanation for the phenomenon of overestimation caused by distribution shift in this section. For a rigorous proof, we refer readers to the related works.

In offline RL, a key challenge arises due to the distribution mismatch between the behavior policy—the policy responsible for generating the data—and the target policy, which is the policy one aims to improve. This mismatch can result in extrapolation errors during value function estimation, often leading to overestimation. Specifically, during the policy evaluation stage, the dataset may not encompass all possible state-action pairs in the Markov Decision Process (MDP), leading to inaccurate $Q$ function estimates for unseen state-action pairs. These estimates may be either too high or too low compared to the actual $Q$ values. Subsequently, in the policy improvement stage, the algorithm tends to shift towards actions that appear to offer higher rewards based on these overestimated values. Unlike online RL, which can implement the current training policy in the environment to obtain real feedback and thereby correct the policy's direction, offline RL lacks this corrective mechanism unless careful loss design is employed. When using common bootstrapping methods like temporal difference learning for training the $Q$ function, these overestimation errors can propagate, affecting estimates

for other state-action pairs. This can result in a chain reaction of overestimations, potentially causing an exponential explosion of the $Q$ function.

