$$\pi^*_{MA}(a|s) \leftarrow \arg\max_{\pi} J(\pi, \hat{M}) - \alpha \frac{1}{1-\gamma} \mathbb{E}_{s \sim d^{\pi}_{\hat{M}}(s)}[D_{CQL}(\pi, \beta)(s)]. \tag{A.21}$$

 Recall that $\forall s, \pi, \beta, D_{CQL}(\pi, \beta)(s) \geq 0$. We have

$$
\begin{aligned}
J(\pi^*_{MA}, \hat{M}) \geq & J(\pi^*_{MA}, \hat{M}) - \alpha \frac{1}{1-\gamma} \mathbb{E}_{s \sim d^{\pi^*_{MA}}_{\hat{M}}(s)}[D_{CQL}(\pi^*_{MA}, \beta)(s)] \\
\geq & J(\pi^*, \hat{M}) - \alpha \frac{1}{1-\gamma} \mathbb{E}_{s \sim d^{\pi^*}_{\hat{M}}(s)}[D_{CQL}(\pi^*, \beta)(s)].
\end{aligned}
\tag{A.22}
$$

 Then we give an upper bound of $\mathbb{E}_{s \sim d^{\pi^*}_{\hat{M}}(s)}[D_{CQL}(\pi^*, \beta)(s)]$. Due to the assumption that $\beta^i$ is
 greater than $\epsilon$ anywhere, we have

$$
\begin{aligned}
D_{CQL}(\pi, \beta)(s) = & \sum_a \pi(a|s)[\frac{\pi(a|s)}{\beta(a|s)} - 1] = \sum_a \pi(a|s)[\frac{\pi(a|s)}{\prod_{i=1}^n \beta^i(a^i|s)} - 1] \\
\leq & \left( \frac{1}{\epsilon^n} \sum_a \pi(a|s)[\pi(a|s)] \right) - 1 \leq \frac{1}{\epsilon^n} - 1.
\end{aligned}
\tag{A.23}
$$

 Combining Eq. A.22 and Eq. A.23, we can get

$$J(\pi^*_{MA}, \hat{M}) \geq J(\pi^*, \hat{M}) - \frac{\alpha}{1-\gamma}(\frac{1}{\epsilon^n} - 1) \tag{A.24}$$

 Recall the sampling error proved in [4] and referred to above in (A.10), we can use it to bound the
 performance difference for any $\pi$ on true and empirical MDP by

$$|J(\pi, M) - J(\pi, \hat{M})| \leq \frac{C_{r,T,\delta} R_{max}}{(1-\gamma)^2} \sum_s \frac{\rho(s)}{\sqrt{|D(s)|}}, \tag{A.25}$$

 then let $sampling\ error := 2 \cdot \frac{C_{r,T,\sigma} R_{max}}{(1-\gamma)^2} \sum_s \frac{\rho(s)}{\sqrt{|D(s)|}}$, and incorporate it into (A.24) , we get

$$J(\pi^*_{MA}, M) \geq J(\pi^*, M) - \frac{\alpha}{1-\gamma}(\frac{1}{\epsilon^n} - 1) - sampling\ error \tag{A.26}$$

 where $sampling\ error$ is a constant dependent on the MDP itself and D. Note that during the proof
 we do not take advantage of the nature of $\pi^*$. Actually $\pi^*$ can be replaced by any policy $\pi$. The
 reason we use $\pi^*$ is that it can give that largest lower bound, resulting in the best policy improvement
 guarantee. Similarly, $D^{CF}_{CQL}$ can be bounded by $\frac{1}{\epsilon} - 1$:

$$
\begin{aligned}
D^{CF}_{CQL}(\pi, \beta)(s) = & \sum_{i=1}^n \lambda_i \sum_{a^i} \pi^i(a^i|s)[\frac{\pi^i(a^i|s)}{\beta^i(a^i|s)} - 1] \\
\leq & \left( \frac{1}{\epsilon} \sum_{i=1}^n \lambda_i \sum_{a^i} \pi^i(a^i|s)[\pi^i(a^i|s)] \right) - 1 \\
\leq & \frac{1}{\epsilon} \left( \sum_{i=1}^n \lambda_i \right) - 1 = \frac{1}{\epsilon} - 1.
\end{aligned}
\tag{A.27}
$$

 $\qquad\square$

## A.5 Proof of Theorem 4.4

We first show the theorem of safe policy improvement guarantee for MACQL and CFCQL, separately. Then we compare these two gaps.

MACQL has a safe policy improvement guarantee related to the number of agents $n$:

**Theorem A.1.** *Given the discounted marginal state-distribution $d_{\hat{M}}^{\boldsymbol{\pi}}$, we define $\mathcal{B}(\boldsymbol{\pi}, D) = \mathbb{E}_{s \sim d_{\hat{M}}^{\boldsymbol{\pi}}}[\sqrt{D(\boldsymbol{\pi}, \boldsymbol{\beta})(s) + 1}]$. The policy $\boldsymbol{\pi}_{MA}^*(\boldsymbol{a}|s)$ is a $\zeta^{MA}$-safe policy improvement over $\boldsymbol{\beta}$ in the actual MDP $M$, i.e., $J(\boldsymbol{\pi}_{MA}^*, M) \geq J(\boldsymbol{\beta}, M) - \zeta^{MA}$, where $\zeta^{MA} = 2\left(\frac{C_{r,\delta}}{1-\gamma} + \frac{\gamma R_{\max} C_{T,\delta}}{(1-\gamma)^2}\right) \cdot \frac{\sqrt{|A|}}{\sqrt{|\mathcal{D}(s)|}} \mathcal{B}(\boldsymbol{\pi}_{MA}^*, D_{CQL}) + \frac{\alpha}{1-\gamma}(\frac{1}{\epsilon^n} - 1) - (J(\boldsymbol{\pi}^*, \hat{M}) - J(\hat{\boldsymbol{\beta}}, \hat{M}))$.*

*Proof.* We can first get a $J(\boldsymbol{\pi}_{MA}^*, \hat{M})$-related policy improvement guarantee following the proof of Theorem 3.6 in Kumar et al. [4]:

$$
\begin{aligned}
J(\boldsymbol{\pi}_{MA}^*, M) \geq & J(\boldsymbol{\beta}, M) - \left(2\left(\frac{C_{r,\delta}}{1-\gamma} + \frac{\gamma R_{\max} C_{T,\delta}}{(1-\gamma)^2}\right) \cdot \frac{\sqrt{|A|}}{\sqrt{|\mathcal{D}(s)|}} \mathcal{B}(\boldsymbol{\pi}_{MA}^*, D_{CQL}) \right. \\
& \left. - (J(\boldsymbol{\pi}_{MA}^*, \hat{M}) - J(\hat{\boldsymbol{\beta}}, \hat{M}))\right)
\end{aligned}
\tag{A.28}
$$

According to Eq. A.21, $\boldsymbol{\pi}_{MA}^*$ is obtained by optimizing $J(\boldsymbol{\pi}, \hat{M})$ with a $D_{CQL}$-related regularizer. And Theorem 4.3 shows that $D_{CQL}$ can be extremely large when the team size expands, which may severely change the optimization objective and affects the shape of the optimization plane. Therefore, $J(\boldsymbol{\pi}_{MA}^*, \hat{M})$ may be extremely low, and keeping $J(\boldsymbol{\pi}_{MA}^*, \hat{M})$ in Eq. A.28 results in a mediocre policy improvement guarantee. To bound $J(\boldsymbol{\pi}_{MA}^*, \hat{M})$, we introduce Eq. A.24 into Eq. A.28, we get the following:

$$
\begin{aligned}
J(\boldsymbol{\pi}_{MA}^*, M) \geq & J(\boldsymbol{\beta}, M) - \left(2\left(\frac{C_{r,\delta}}{1-\gamma} + \frac{\gamma R_{\max} C_{T,\delta}}{(1-\gamma)^2}\right) \cdot \frac{\sqrt{|A|}}{\sqrt{|\mathcal{D}(s)|}} \mathcal{B}(\boldsymbol{\pi}_{MA}^*, D_{CQL}) \right. \\
& \left. + \frac{\alpha}{1-\gamma}(\frac{1}{\epsilon^n} - 1) - (J(\boldsymbol{\pi}^*, \hat{M}) - J(\hat{\boldsymbol{\beta}}, \hat{M}))\right)
\end{aligned}
\tag{A.29}
$$

This complete the proof. $\qquad\square$

We can get a similar $\zeta^{CF}$ satisfying $J(\boldsymbol{\pi}_{CF}^*, M) \geq J(\boldsymbol{\beta}, M) - \zeta^{CF}$ for CFCQL, which is independent of $n$:

$$
\zeta^{CF} = 2\left(\frac{C_{r,\delta}}{1-\gamma} + \frac{\gamma R_{\max} C_{T,\delta}}{(1-\gamma)^2}\right) \cdot \frac{\sqrt{|A|}}{\sqrt{|\mathcal{D}(s)|}} \mathcal{B}(\boldsymbol{\pi}_{CF}^*, D_{CQL}^{CF}) + \frac{\alpha}{1-\gamma}(\frac{1}{\epsilon} - 1) - (J(\boldsymbol{\pi}^*, \hat{M}) - J(\hat{\boldsymbol{\beta}}, \hat{M}))
\tag{A.30}
$$

Then we can prove Theorem 4.4.

*Proof.* Subtract $\zeta^{CF}$ from $\zeta^{MA}$, and we get:

$$
\zeta^{MA} - \zeta^{CF} = 2\left(\frac{C_{r,\delta}}{1-\gamma} + \frac{\gamma R_{\max} C_{T,\delta}}{(1-\gamma)^2}\right) \frac{\sqrt{|A|}}{\sqrt{|\mathcal{D}(s)|}}\left(\mathcal{B}(\boldsymbol{\pi}_{MA}^*, D_{CQL}) - \mathcal{B}(\boldsymbol{\pi}_{CF}^*, D_{CQL}^{CF})\right) + \frac{\alpha}{1-\gamma}(\frac{1}{\epsilon^n} - \frac{1}{\epsilon})
\tag{A.31}
$$

Let the right side $\geq 0$, and we can get

$$
n \geq \log_{\frac{1}{\epsilon}}\left[\max\left(1, \frac{1}{\epsilon} + \frac{2}{\alpha}\frac{\sqrt{|A|}}{\sqrt{|\mathcal{D}(s)|}}\left(C_{r,\delta} + \frac{\gamma R_{\max} C_{T,\delta}}{1-\gamma}\right) \cdot \left[\mathcal{B}\left(\boldsymbol{\pi}_{CF}^*, D_{CQL}^{CF}\right) - \mathcal{B}\left(\boldsymbol{\pi}_{MA}^*, D_{CQL}\right)\right]\right)\right]
\tag{A.32}
$$

According to Theorem 4.3,

$$
\mathcal{B}\left(\boldsymbol{\pi}_{CF}^*, D_{CQL}^{CF}\right) = \mathbb{E}_{s \sim d_{\hat{M}}^{\boldsymbol{\pi}_{CF}^*}}[\sqrt{D_{CQL}^{CF}(\boldsymbol{\pi}_{CF}^*, \boldsymbol{\beta})(s) + 1}] \leq \mathbb{E}_{s \sim d_{\hat{M}}^{\boldsymbol{\pi}_{CF}^*}}[\sqrt{\frac{1}{\epsilon} - 1 + 1}] = \frac{1}{\sqrt{\epsilon}}
\tag{A.33}
$$

In the meantime, we have

$$\mathcal{B}\left(\boldsymbol{\pi}_{CF}^*, D_{CQL}^{CF}\right) = \mathbb{E}_{s \sim d_{\hat{M}}^{\boldsymbol{\pi}_{MA}^*}}[\sqrt{D_{CQL}(\boldsymbol{\pi}_{MA}^*, \boldsymbol{\beta})(s) + 1}] \ge \mathbb{E}_{s \sim d_{\hat{M}}^{\boldsymbol{\pi}_{MA}^*}}[\sqrt{D_{CQL}(\boldsymbol{\beta}, \boldsymbol{\beta})(s) + 1}] = 1$$
(A.34)

Therefore, we can relax the lower bound of $n$ to a constant that

$$n \ge \log_{\frac{1}{\epsilon}}\left(\frac{1}{\epsilon} + \frac{2}{\alpha}\frac{\sqrt{|A|}}{\sqrt{|\mathcal{D}(s)|}}(C_{r,\delta} + \frac{\gamma R_{\max} C_{T,\delta}}{1-\gamma}) \cdot (\frac{1}{\sqrt{\epsilon}} - 1)\right)$$
(A.35)

$\square$

# B   Implement Details

## B.1   Derivation of the Update Rule

To utilize the Eq. 4 for policy optimization, following the analysis in the Section 3.2 in Kumar et al. [4], we formally define optimization problems over each $\mu^i(a^i|s)$ by adding a regularizer $R(\mu^i)$. As shown below, we mark the modifications from the Eq. 4 in red.

$$\min_Q \max_{\boldsymbol{\mu}} \alpha\left[\sum_{i=1}^n \lambda_i \mathbb{E}_{s \sim \mathcal{D}, a^i \sim \mu^i, \boldsymbol{a}^{-i} \sim \boldsymbol{\beta}^{-i}}[Q(s, \boldsymbol{a})] - \mathbb{E}_{s \sim \mathcal{D}, \boldsymbol{a} \sim \boldsymbol{\beta}}[Q(s, \boldsymbol{a})]\right]$$
$$+ \frac{1}{2}\mathbb{E}_{s, \boldsymbol{a}, s' \sim \mathcal{D}}\left[(Q(s, \boldsymbol{a}) - \hat{\mathcal{T}}^{\boldsymbol{\pi}}\bar{Q}_k(s, \boldsymbol{a}))^2\right] + \sum_{i=1}^n \lambda_i R(\mu^i),$$
(B.36)

By choosing different regularizer, there are a variety of instances within CQL family. As recommended in Kumar et al. [4], we choose $R(\mu^i)$ to be the KL-divergence against a Uniform distribution over action space, i.e., $R(\mu^i) = -D_{KL}(\mu^i, Unif(a^i))$, then it's easily to derive the following variant of Eq. B.36 called CFCQL($H$) which is the update rule we used: