# OpenReview forum: "Counterfactual Conservative Q Learning for Offline Multi-agent Reinforcement Learning"
_NeurIPS.cc/2023/Conference — NeurIPS 2023 poster_

### Official Review · Reviewer_EZbv · 2023-07-03

**Soundness:** 3 good
**Presentation:** 2 fair
**Contribution:** 4 excellent
**Rating:** 7
**Confidence:** 3

**Summary:**

The paper addresses the problem of offline multi-agent reinforcement learning. A way to make conservative updates to the Q-function is proposed, extending, non-trivially, CQL to multi-agent settings. Theory and experiments clearly validate the approach.

**Strengths:**

The method is both well motivated as well as theoretically and empirically validated. The method is experimentally shown to be substantially better than all the state-of-the-art baselines considered.

**Weaknesses:**

The introduction can be improved in terms of written language. Moreover, I would like to see a couple of straightforward independent learning baselines added to the StarCraft results. See my questions below.

Minor comments:
- I do not understand the last sentence of 3.3;
- Some sentences to improve: figure 1(a), first sentence; first paragraph of the introduction;
- I suggest giving an intuitive meaning to Theorems 1 and 2 after they appear, as well as an intuitive meaning of the whole theory of Section 4.3 in the end of the Section.
- typos: "temporature" in the caption of Figure 3(a); capitalize "conclusion" in the title of Section 6.

**Questions:**

- In Section 4.1, what are the optimal Q-values of the Toy MDP ?
- What is $\beta$ ? It appears first in Section 3.3 and I don't see it defined, even though it seems quite important in the sequel (for instance, it appears twice in the proposed algorithm).
- Why is the method called counterfactual ? I suggest adding an intuitive explanation to the document.
- What is the performance of single agent independent Q-learning (actually DQN) and of independent CQL (actually DQN) on the SMAC environment ? I believe it would be quite important to make clear on the table of the results that coordination is really necessary to perform the task and that ICQL is not enough.

**Limitations:**

I can not see the limitations of the work nor I foresee a negative societal impact.

---

> ### Author Rebuttal · Authors · 2023-08-08
>
> Thank you for your valuable comment!
>
> ●**Q1: Typos, sentence improvement, and intuition explanation.**
>
> As per your suggestion, we have fixed the typos and reorganized the sentences. The intuition behind our theories is as follows: Theorems 4.1 and 4.2 in our paper illustrate the differences in the extent of underestimation of value functions between CFCQL and MACQL. When MACQL samples $n$ OOD actions, it leads to significantly larger gaps between estimated V and true V compared to CFCQL. Additionally, Theorem 4.4 demonstrates that a greater gap between value functions leads to a larger discrepancy between the performance of the trained policy and the behavior policy. These updates and explanations will be reflected in the revised script.
>
> ●**Q2: The explanation of the last sentence of 3.3.**
>
> We apologize for the unclear statement. We want to convey that with a large enough hyper-parameter $\alpha$ in Eq. 2 as the training loss, we can obtain a  $Q$ function, and the corresponding value function, which is lower than the real value function $V^{\pi}(s)=E_{\pi}[\sum_{i=0}^{\infty}\gamma^i r_i]$ on the same MDP at any state. Based on this, the overestimation problem in single agent RL can be resolved.
>
> ●**Q3: In Section 4.1, what are the optimal Q-values of the Toy MDP?**
>
> The Multi-Agent Markov Decision Process (MMDP) proposed in Section 4.1 is an infinite Markov Decision Process (MDP) task, wherein the maximum reward attainable at each step is 1, contingent upon all agents reaching and remaining in state S2. The discount factor (\gamma) was set to 0.8, and consequently, the optimal Q-value is determined to be 1/(1-0.8)=5, as illustrated in Fig. 1(b) with a dashed line.
>
> ●**Q4: What is $\beta$? It appears first in Section 3.3 and I don't see it defined, even though it seems quite important in the sequel (for instance, it appears twice in the proposed algorithm).**
>
> We sincerely apologize for the unclear definition in our paper and any resulting confusion it may have caused.  In our work, we denote $\beta$ as the behavior policy responsible for generating the dataset D. It is important to note that $\beta$ is often represented as an implicit distribution, and in our proposed algorithm, we resort to employing an empirical policy approximation through the behavior-cloning method.
>
> ●**Q5: Why the method is called counterfactual? I suggest adding an intuitive explanation to the document.**
>
> We refer to our method as 'counterfactual' because its structure bears resemblance to counterfactual methods commonly used in multi-agent reinforcement learning (MARL). This involves obtaining each agent's counterfactual baseline by marginalizing out a single agent's action while keeping the other agents' actions fixed. The intuitive rationale behind employing a counterfactual-like approach is that by individually penalizing each agent's out-of-distribution (OOD) actions while holding the other agents' actions constant from the datasets, we can effectively mitigate the out-of-distribution problem in offline MARL with reduced pessimism. As per your suggestion, we will add the explanation to the revised version.
>
> ●**Q6: What is the performance of single agent independent Q-learning (actually DQN) and of independent CQL on the SMAC environment?**
>
> We conducted additional experiments to demonstrate the performance of independent Q-learning (IDQN) and independent CQL (ICQL) on the SMAC environment as listed below. As expected, IDQN proved to be inadequate in handling the offline tasks due to the absence of a specialized mechanism to address the distribution shift issue. ICQL demonstrated promising performance, primarily attributed to its pessimistic design, yet it fell short in achieving high-level performance due to the limitations in coordination imposed by the independent learning paradigm. The supplementary experiments underscore the essentiality of integrating pessimism and CTDE paradigm within the offline MARL domain.
>
> | Map | Dataset | CFCQL| IDQN | ICQL |
> | ------- | --------- | ------- | --------- | ---------------- |
> |5m_vs_6m|Medium| **0.29±0.05** |0.00±0.00| 0.21±0.04|
> ||Medium_Replay|**0.22±0.06**|0.00±0.00 | 0.21±0.03|
> ||Expert|**0.84±0.03**|0.00±0.00| 0.73±0.06|
> ||Mixed|**0.76±0.07** |0.00±0.00| 0.72±0.07|
> |6h_vs_8z|Medium| **0.41±0.04**|0.00±0.00| 0.35±0.04|
> ||Medium_Replay|**0.21±0.05**|0.00±0.00 | 0.09±0.05|
> ||Expert|**0.7±0.06**|0.00±0.00| 0.49±0.10|
> ||Mixed|**0.49±0.08** |0.00±0.00|0.33±0.06 |
>
> ●**Q7: Lack of discussions on limitations and broader impacts:**
>
> Due to the page limitation, we have put these to Appendix E.

---

> > ### Comment · Reviewer_EZbv · 2023-08-16
> > **Acknowledgment of rebuttal**
> >
> > I thank the author for their response. In light of the answers and the the author discussion with other reviewers, I am keeping my score.

---

### Official Review · Reviewer_dhyd · 2023-07-05

**Soundness:** 2 fair
**Presentation:** 1 poor
**Contribution:** 2 fair
**Rating:** 4
**Confidence:** 4

**Summary:**

Offline Reinforcement Learning in the multi-agent setting suffers from the combined effects of distribution shift and increasing number of agents. An exponential blowup of the action space in addition to Out-Of Distribution (OOD) actions hinders performance of RL agents. The work tackles these phenomena by proposing CounterFactual Conservative Q Learning (CFCQL), an algorithm that learns conservatively from static offline datasets. CFCQL differs from the naive multi-agent CQL approach as it conservatively trains each agent separately using the pessimistic regularization. Regularization terms are then linearly combined for global value estimation. This prevents agents from being excessively conservative while maintaining theoretical guarantees of underestimation and safe policy improvement as in CQL. In practice, weighted contributions of conservative penalties are realized either using a one-hot encoding or softmax with temperature scaling. In the case of continuous actions, counterfactual Q function updates of the gradient are utilized. Experiments are carried out on a range of discrete and continuous multi-agent benchmarks with promising improvements.

**Strengths:**

* The paper presents an amenable combination of design choices.
* Empirical results are promising and extensive.

**Weaknesses:**

* **Writing and Presentation**: My main concern is the writing and presentation of the paper. The paper is not well written and presented in an unorganized manner. Technical claims made by authors are vague and informal. Explanations are not well supported by intuition or insights and the frequency of grammatical errors is too high. Specifically, sections 1 and 2 motivate the work with high-level and vague statements. Section 3.3 does not formally explain the offline RL problem (of maximizing the expected discounted return), behavior policy or distribution shift. Section 4 builds the algorithm using informal vocabulary. Section 4.4 presents theoretical guarantees without any intuition or interpretation and section 6 summarizes the paper informally. In my view, the paper's presentation requires significant attention.
* **Dataset Ablations**: I am having trouble understanding results on dataset ablations presented in figure 3 (b). Ideally, as the number of data samples grow, the performance of agents improves. With larger static datasets agents have access to more in-distribution samples and a broader coverage of the underlying MDP. However, figure 3 (b) demonstrates that performance of agents drops for 50,000 samples. It would be helpful if the authors can explain this result or its interpretation. In its current form, it appears that the approach may not scale well to larger multi-agent learning datasets for real-world applications.
* **Baselines**: While the paper includes relevant multi-agent learning baselines, it is worth noting that none of these baselines were designed for offline RL. All methods were developed as off-policy or pure online RL methods that aggregate new experience. With that said, the only baseline of interest is MACQL since it leverages the CQL penalty designed for static datasets. Authors should consider offline-RL baselines that may be adapted for multi-agent learning. For instance, IQL[1] and TD3-BC[2] are state-of-the-art offline RL methods which might be useful for comparison. Similarly, AWR[3] is another method that imitates dataset interactions.
* **Transfer and Finetuning**: The paper claims that CFCQL addresses distribution shift and generalizes better. However, this claim is not well supported. Ideally, robustness and generalization ability of an algorithm are tested by transferring it to new unseen scenarios. Authors should consider finetuning agents or adapting them to a new task even if on a small toy example. This will help verify the claims of CFCQL addressing distribution shift.


While experiments and results are promising, overall presentation and writing of the paper needs significant improvement.

## Minors

* line 17: man-made -> synthetic
* line 23: ~from~
* line 24: ~the~
* line 24: highly -> high
* line 37: agent number -> number of agents
* line 44: ~just~
* line 55: bounded from below -> lower bounded
* line 63: too much -> excessive
* line 66: contributes CQL -> contributes to CQL
* line 69: sampled in the dataset -> sampled in-distribution
* line 70: agents number -> number of agents
* line 72: man-made -> synthetic
* line 79: advantagous -> advantageous
* line 149: dataset distribution -> behavior policy
* Figure 1: stays -> stay
* line 171: even worse -> more significantly
* line 175: need learn -> need to learn
* line 178: exponentially -> exponential
* line 183: style -> update
* line 198: contributes regularization -> contributes its regularization
* line 201: in the dataset -> in-distribution
* line 238: performances -> performance
* line 257: contributes penalty -> contributes to penalty
* line 258: style -> encoding
* line 307: agents number -> number of agents
* line 314: basically -> mostly
* Figure 3: temporature -> temperature
* line 372: conlcusion -> Conclusion
* line 373: lack of -> lacks
* line 374: from theories -> using theory
* line 375: exponentially -> exponential
* line 379: exponentially -> exponential
* line 381: achieve -> achieves
* line 381: Some -> Ablation studies
* line 382: also made -> conducted

[1]. Kostrikov et. al., Offline Reinforcement Learning with Implicit Q-Learning, ICLR 2022.
[2]. Scott Fujimoto, Shixiang Shane Gu, A Minimalist Approach to Offline Reinforcement Learning, NeurIPS 2021.
[3]. Peng et. al., Advantage Weighted Regression: Simple and Scalable Off-Policy Reinforcement Learning, arXiv 2020.

**Questions:**

* What is the formal problem definition of offline RL? What is a behavior policy? What is distribution shift?
* What is the interpretation of Figure 3 (b)? How does an increase in the size of dataset lead to a decrease in agent performance? How does CFCQL scale for larger data samples?
* Can you please explain the reasoning behind off-policy/online RL methods being relevant baselines? Can CFCQL be compared to IQL, TD3-BC or AWR even if on small toy tasks?
* How does CFCQL address distribution shift? How does CFCQL generalize to new task? Can CFCQL be adapted to new multi-agent tasks or transferred to different kinds of agents following pretraining?

**Limitations:**

Authors have discussed limitations and future work.

---

> ### Author Rebuttal · Authors · 2023-08-08
>
> Thank you for your thoughtful comments and detailed syntax error check!
>
> ●**Q1: Presentation problem. What is the formal problem definition of offline RL? What is a behavior policy? What is distribution shift?**
>
> We apologize for the unclear statement, and we are committed to improving the presentation. To achieve this, we will seek assistance from native speakers.
>
> As stated in Section 3.1, the online MARL problem concerned in this work is formulated in the context Partially Observable Markov Decision Process (POMDP). And the target is to maximize the cumulative reward $E_{\pi}[\sum_{t=0}^{\infty}\gamma^t r(s_t, a_t)]$. When transferred to offline MARL, the primary distinction is that the training data is a static dataset of transitions, denoted by $D={(s_t,o^i_t, a^i_t,s_{t+1},o^i_{t+1}, r^i_t)}$, where $i\le n$ means the agent index, and $t$ denotes the time-step. Additionally, the agent no longer has the ability to collect data by interacting with the environment.
>
> The behavior policy $\beta$ denotes the distribution over states and actions in D and the transitions in D could be interpreted as being sampled according to the behavior policy, i.e., $s\sim d^\beta(s)$, $a\sim \beta(a|s)$, where $d^\beta(s)$ means the state visitation distribution under policy $\beta$.
>
> Offline RL algorithms encounter the distribution shift issue. The distribution of the behavior policy generally differs from the distribution of the learned policy. Consequently, when updating the Q-values with Bellman backups, the actions sampled from the learned policy may be out-of-distribution (OOD) actions, potentially leading to the overestimation of Q-values[1]. Due to the inability to interact with the environment to correct such errors, the distribution shift issue becomes a major challenge in offline RL.
>
> ●**Q2: What is the interpretation of Figure 3 (b)? How does an increase in the size of dataset lead to a decrease in agent performance? How does CFCQL scale for larger data samples?**
>
> The decrease in the performance of larger datasets (50k trajectories) in Figure 3 (b) occurs because of the under-fitting issue. To ensure fairness, the maximum number of training steps for all datasets and algorithms on the SMAC environment is fixed at $1\times 10^7$. However, this may result in termination of trainings before convergence is achieved, resulting in the under-fitting issue.
>
> To verify our claim aforementioned, we retrained the CFCQL on datasets containing 5,000 or 50,000 trajectories until convergence. The performances of the converged agents and the steps taken to achieve convergence are presented in the table below. The results demonstrate that larger datasets contribute to improved convergence performances, thus confirming the scalability of CFCQL for larger data samples.
> |Performance|Medium|Expert|Mixed|
> |-----------|--------|--------|---------|
> |5k(showed in the paper)|0.41±0.04|0.7±0.06|0.49±0.08|
> |50k(showed in the paper)|0.36±0.13|0.63±0.09|0.5±0.13|
> |5k(converged)|0.43±0.07|0.75±0.05|0.65±0.08|
> |50k(converged)|**0.47±0.07**|**0.79±0.07**|**0.67±0.07**|
>
> |Convergence Steps($\times 10^7$)|Medium|Expert|Mixed|
> |-----------|--------|--------|---------|
> |5k|2|2|2|
> |50k|2|2|5|
>
> Regarding the Medium-Replay dataset, we have discovered a bug in our code and the maximum number of trajectories in the Medium-Replay datasets is 5000 instead of 50000. Since only 5000 trajectories are generated when agents are trained to medium performance, we cannot fill a replay buffer with 50000 trajectories for this kind of dataset. Therefore, the training results pertaining to Medium-Replay datasets with 5,0000 trajectories are invalid. We deeply apologize for this mistake and have rectified it in our code. We will revise our paper accordingly.
>
> ●**Q3: How does CFCQL address distribution shift? How does CFCQL generalize to new task?**
>
> We acknowledge that the distribution shift problem in offline RL is distinct from the generalization problem. In offline RL, the distribution shift refers to the divergence between the trajectory distribution generated by the behavior policy $\beta$ (i.e., the dataset) and the imagined distribution of trajectories generated by the current training policy $\pi$. The main challenge in offline RL is to provide accurate feedback to the current policy $\pi$ using the data distribution induced by another policy $\beta$, which constitutes the distribution shift problem. We tackle this problem with a counterfactual regularizer to penalize the action that is not sampled from the dataset, i.e., the left part of Eq. 4 in the original paper.
>
> In our work, both the dataset and our trained policy share the same task and reward function, and we do not require agents to possess generalization abilities for transferring to a new task. Instead, we focus on addressing the distribution shift problem to effectively learn from offline data and improve policy performance within the given task.
>
> ●**Q4: Can you please explain the reasoning behind off-policy/online RL methods being relevant baselines? Can CFCQL be compared to IQL, TD3-BC or AWR even if on small toy tasks?**
>
> Due to the page limit, please refer to the **public rebuttal block** with name "Author Rebuttal by Authors" on the top of this page for the explanation of lacking baselines.
>
> Reference:
>
> [1] Fujimoto, Scott, David Meger, and Doina Precup. "Off-policy deep reinforcement learning without exploration." International conference on machine learning. PMLR, 2019.

---

> > ### Comment · Reviewer_dhyd · 2023-08-13
> > **Response to Authors' Comments**
> >
> > I thank the authors for providing a detailed response. After going through authors responses and other reviewers' comments, my concerns regarding dataset ablations still remain unaddressed.
> >
> > * **Dataset Ablations**- I am struggling to understand the under-fitting issue of CFCQL towards the behavior policy. Assuming the policy is trained for $10^7$ steps, 50k samples still present a broader coverage of the MDP for the same policy. Intuitively, if the behavior policy is kept same and the number of samples are increased, the data distribution remains unchanged and does not affect performance of the downstream agent. Irrespective of underfitting, the CQL algorithm is known to find performant policies even from sub-optimal unconverged behavior policies [1, 2]. Thank you for bringing up the `medium-replay` dataset composition. From my understanding, `medium` datasets can be filled up to 50k samples by simply letting the policy run in the environment and collecting samples from different seeds. Note that in this setting we do not make gradient updates to the policy. Nevertheless, it is worth looking into the dataset composition and exact counts of samples in each task dataset.
> >
> > [1]. Kumar et. al., A Workflow for Offline Model-Free Robotic Reinforcement Learning, CoRL 2021.
> >
> > [2]. Kumar et. al., Should I Run Offline Reinforcement Learning or Behavioral Cloning?, ICLR 2022.

---

> > > ### Author Response · Authors · 2023-08-14
> > > **Response to Your Comments**
> > >
> > > We appreciate your feedback and apologize for the misleading aspects of our rebuttal. Firstly, we would like to clarify that the “underfitting” issue is directed at the training policy $\pi$, not the behavior policy $\beta$. The “$1\times 10^7$ training steps” indicate that we train the downstream agents for $1\times 10^7$ steps on each dataset, as described in the original paper. As demonstrated in the previous rebuttal section, $1\times 10^7$ steps are insufficient for the downstream agents to converge (see the second table in the previous rebuttal section). When trained to convergence, the policy trained on a 50k dataset slightly outperforms the policy trained on a 5k dataset (refer to the first table in the previous rebuttal section). These findings align with your expectation in the first comment that, "Ideally, as the number of data samples increase, the performance of agents should improve."
> > >
> > > Regarding the medium dataset issue, we have reviewed its composition and can confirm it aligns with your suggestion to "let the policy run in the environment and collect samples from various seeds." If any aspect of our statement remains unclear, please provide further feedback, and we will respond promptly.

---

> > > > ### Comment · Reviewer_dhyd · 2023-08-17
> > > > **Follow Up to Authors' Response**
> > > >
> > > > Thank you for providing details on the underfitting issue and its relation with the dataset composition. However, I would still like to better understand the dataset structure for Equal-Line and StarCraft II experiments. Appendices C.1 and C.2 provide a high-level picture but I have a few specific questions which are listed below-
> > > >
> > > > 1. Could you please provide the exact dataset composition for each type of policy (i.e- `medium`, `medium-replay` and `mixed`)? For instance, how many trajectories are present in the dataset? How many transition samples are present in each trajectory? For how many steps was the behavior policy trained? And how was the data collected (i.e- during training, acting or simply from replay buffer)?
> > > > 2. Could you please provide details on the training procedure? For instance, for how many steps was a CFCQL policy trained using a given dataset? How many random seeds were used for experiments?
> > > > 3. Following my concern on dataset ablations, authors mention that CFCQL often underfits the behavior policy for a given number of steps. While this is not a negative effect of the algorithm, it is still an area of concern since the convergence of CFCQL is a matter of chance for larger datasets. Is the underfitting phenomenon prevalent over random seeds? If yes, then CFCQL may require additional tuning and design considerations.

---

> > > > > ### Author Response · Authors · 2023-08-18
> > > > > **Response to Reviewer dhyd**
> > > > >
> > > > > **1. Exact dataset composition.**
> > > > >
> > > > > We summarize the exact dataset composition as listed below.
> > > > > | Datasets Composition| Trajectories | Transition Samples (Episode Limit) | Training Steps for Sampling Policy|
> > > > > |---|---|---|---|
> > > > > |Equal_Line|1000|50|5e6|
> > > > > |Medium(6h_vs_8z)|5000|150|3e6|
> > > > > |Medium-Replay(6h_vs_8z)|5000|150|3e6|
> > > > > |Expert(6h_vs_8z)|5000|150|1e7|
> > > > > |Mixed(6h_vs_8z)|5000|150|-|
> > > > >
> > > > > It is worth noting that the datasets used in the Equal_Line experiment are all Expert datasets, as discussed in Section 5.1. In addition, the datasets for various maps in the StarCraft II experiment have different compositions, including Transition Samples and Training Steps for Sampling Policy. This variation arises due to the diverse characteristics of the maps. Here, we use the map 6h_vs_8z as an illustrative example for conciseness.
> > > > >
> > > > > We adopt the widely accepted data collection method proposed in D4RL[1], and here are the details.
> > > > > **Medium**: Firstly, we train a QMIX policy until it reaches medium performance, specifically a winning rate of 0.2-0.3 in the SC2 experiment. Then, we execute this partially-pretrained policy in the environment to sample and collect data for the creation of Medium dataset.
> > > > > **Medium-Replay**: Firstly, we train a QMIX policy until it reaches medium performance, specifically a winning rate of 0.2-0.3 in the StarCraft II experiment. Then, we save all trajectories from current replay buffer into Medium-Replay dataset.
> > > > > **Expert**: Firstly, we train a QMIX policy until it converges, specifically 1e7 training steps in the StarCraft II experiment. Then, we execute this fully-pretrained policy in the environment to sample and collect data for the creation of Expert dataset.
> > > > > **Mixed**: After collecting the Medium dataset and Expert dataset, we combine an equal portion from each dataset to create the Mixed dataset.
> > > > >
> > > > > **2. Details on the training procedure.**
> > > > >
> > > > > In the original paper, due to our limited resources, we typically train a CFCQL policy or other baseline policies for 1e7 training steps using any given dataset. However, as demonstrated in the previous sections on rebuttal, additional training steps may be required for convergence.
> > > > > Regarding the random seeds, as stated in Section 5, each algorithm in every experiment is run for five random seeds.
> > > > >
> > > > > **3. Underfitting issue.**
> > > > >
> > > > > The number of convergence steps for each random seed on every dataset is detailed below.
> > > > >
> > > > > |Convergence Steps($\times 10^6$)| seed1 | seed2 |seed3 |seed4|seed5|Mean±Std|
> > > > > |--|--|--|--|--|--|--|
> > > > > |5000(Medium)|11|18|19|18|12|15.6±3.8|
> > > > > |50000(Medium)|14|19|17|19|18|17.4±2.1|
> > > > > |5000(Expert)|19|20|19|17|18|18.6±1.1|
> > > > > |50000(Expert)|17|15|19|20|19|18.0±2.0|
> > > > > |5000(Mixed)|18|20|15|16|19|17.6±2.1|
> > > > > |50000(Mixed)|45|48|45|45|43|45.2±1.8|
> > > > >
> > > > > The training steps needed for convergence remains basically consistent with size 5000 and 50000 except for the mixed dataset, the data distribution of which seems to follow a bimodal distribution between medium policy and expert policy. When juxtaposing with online Reinforcement Learning (RL), offline methods often necessitate more training steps. Take a typical single agent environment MuJoCo  as an example, in online settings, the training step is often set to $5\times10^5$, while in offline settings, it is often set to $1\times10^6$ or even $3\times10^6$[2]. Besides, the efficiency of offline training means it is markedly faster than its online counterpart, which implies that extra training cost is completely acceptable.
> > > > >
> > > > > In our experiments, we typically allocate a maximum of 3 hours for training an offline policy, in contrast to the expert behavior policy which demands at least half a day. Thus, augmenting the number of training steps, up to a certain limit, is a judicious approach and doesn't imply any scalability issues with our algorithm when applied to larger datasets."
> > > > >
> > > > > **Reference:**
> > > > >
> > > > > [1] Fu, J., Kumar, A., Nachum, O., Tucker, G., & Levine, S. (2021). √D4RL: Datasets for Deep Data-Driven Reinforcement Learning (arXiv:2004.07219). arXiv. http://arxiv.org/abs/2004.07219
> > > > >
> > > > > [2] An, Gaon, et al. "Uncertainty-based offline reinforcement learning with diversified q-ensemble." Advances in neural information processing systems 34 (2021): 7436-7447.

---

> > > > > > ### Comment · Reviewer_dhyd · 2023-08-19
> > > > > > **Response to Authors' Comments**
> > > > > >
> > > > > > Thank you for the detailed response consisting of the training setup and details. Below I summarize the current state of my concerns, authors' responses and changes in my opinion during the discussion-
> > > > > >
> > > > > > Since the authors have addressed most of my main concerns around the presentation, choice of baselines and scalability of the proposed algorithm, the overall quality of the paper is improved. Authors provided strong offline RL baselines in the multi-agent setting, promised to improve the writing and presentation of the paper and provided detailed descriptions of their experimental setup. Additionally, authors uncovered the underfitting issue during training which implied a longer training schedule for CFCQL on larger datasets. Unfortunately, the composition of `medium` datasets uncovered during the discussion and its relation with larger dataset sizes prevent me in agreeing with the overall empirical evaluation of CFCQL. It remains unclear as to how CFCQL should be trained on larger datasets and whether the structure and design of datasets is in agreement with literature and training protocols. These issues prevent me from supporting acceptance of the work. Therefore, I would like to slightly raise my score (as a result of the improved paper quality and addressal of my main concerns) and encourage the authors to revisit the dataset design and performance of CFCQL on larger datasets. I thank the authors for their efforts.

---

### Official Review · Reviewer_PGy4 · 2023-07-06

**Soundness:** 3 good
**Presentation:** 2 fair
**Contribution:** 2 fair
**Rating:** 5
**Confidence:** 3

**Summary:**

This paper proposes a novel offline multi-agent reinforcement learning algorithm called Counterfactual Conservative Q-Learning (CFCQL) to address the overestimation issue and achieve team coordination at the same time. The algorithm calculates conservative regularization for each agent separately in a counterfactual way and then linearly combines them to realize an overall conservative value estimation. The paper compares CFCQL and MACQL theoretically and shows that CFCQL is advantageous to MACQL on the performance bounds and safe policy improvement guarantee as the agent number is large. The paper conducts experiments on commonly used multi-agent environments to demonstrate that CFCQL outperforms existing methods on most datasets and even with a remarkable margin on some of them.

**Strengths:**

- Theoretical comparison of the proposed algorithm with existing methods to show its advantages in terms of performance bounds and safe policy improvement guarantee.
- The paper provides a detailed explanation of the proposed algorithm and its counterfactual approach to conservative value estimation.
- Conducting experiments on multiple environments to demonstrate the effectiveness of the proposed algorithm.

**Weaknesses:**

- The paper seems to be a simple combination of CQL and MARL.
- Although the paper shows that the proposed algorithm is theoretically superior to existing methods when the agent number is large, it does not provide empirical evidence to support this claim.

typos
- Section "conclusion" -> "Conclusion"

**Questions:**

- How to get $a_i$ in eq.7? Or what's the range of the summation?
- To my understanding, does CFCQL update $Q$ for each agent's action iteratively instead of jointly in MACQL? In this case, does the order of the updates matter? And why sample only OOD actions from one agent instead of two or more if two agents' action are highly related?

**Limitations:**

The paper does not explicitly mention any limitations of the proposed method. The paper has no potential negative societal impact.

---

> ### Author Rebuttal · Authors · 2023-08-08
>
> Thank you for your insightful feedback!
>
> ●**Q1: No empirical evidence to support the superior of our method on large agent number $n$.**
>
> We apologize for any unclear experimental instructions in our paper. The evidence supporting our claim can be found in Section 5.1, specifically in the demo labeled 'Equal Line'. In Figure 2, it is evident that the performance of MACQL degrades significantly as the number of agents increases, while the performance of CFCQL remains relatively stable. These results strongly support the conclusion we presented in Section 4.3, highlighting that CFCQL provides better policy improvement guarantees compared to MACQL when the number of agents $n$ is sufficiently large.
>
> ●**Q2: The paper seems to be a simple combination of CQL and MARL.**
>
> While CFCQL shares some similarities in loss form with CQL, it differs fundamentally from a simple combination of CQL and MARL (MACQL). Our theoretical analysis demonstrates that MACQL leads to an exponential increase in pessimism as the number of agents increases, resulting in an overly pessimistic value function. To address this issue, a potential solution is to separately penalize each agent's out-of-distribution (OOD) actions, allowing the penalty term to grow linearly rather than exponentially.
> CFCQL successfully adapts the counterfactual method to offline multi-agent settings by independently penalizing each agent's OOD actions while keeping the other agents' actions sampled from datasets. This adaptation holds promise in alleviating the out-of-distribution problem with reduced pessimism. Moreover, the empirical results from our experiments further validate the superiority of CFCQL over MACQL. These unique contributions distinguish CFCQL and demonstrate its effectiveness in multi-agent scenarios.
>
> ●**Q3: How to get $a_i$ in eq.7? Or what's the range of the summation?**
>
> We sincerely apologize for the lack of clarity in our paper and any resulting confusion it may have caused.  The summation range of $a_i$ in Eq. 7 corresponds to the action space of agent $i$, which is similar to the concept in CQL. In terms of implementation details, for the discrete action space, we perform the summation directly over the action space using the standard torch.logsumexp() function, which allows us to efficiently compute the required summation. However, for the continuous action space, we employ an importance sampling method inspired by the CQL. Specifically, we generate action samples from both a uniform-at-random distribution (Unif(a)) and the current policy. These action samples are then utilized in conjunction with importance sampling to compute the summation.
>
> ●**Q4: Does CFCQL update $Q$ for each agent's action iteratively instead of jointly in MACQL? In this case, does the order of the updates matter?**
>
> In CFCQL, the symbol $Q$ refers to the joint state-action function shared by all $n$ agents, which undergoes simultaneous updates involving both the counterfactual terms and the TD error. And we train the agents simultaneously by taking the summation of all $n$ counterfactual terms as the training loss.
>
> ●**Q5: Why sample only OOD actions from one agent instead of two or more if two agents' action are highly related?**
>
> The significance of the number of agent with OOD actions directly influences the disparity between the estimated value functions and the true value functions.  It is possible to sample OOD joint-actions for two agents as MACQL does. However, as illustrated in theorems 4.1 and 4.2 in our paper, this leads to an increased order of pessimism, which is squared compared to CFCQL.  And as more agents are involved, the induced pessimism will explode exponentially. However, as Theorem 4.3 demonstrates, a greater gap between the estimated and true value functions will lead to a larger discrepancy between the performance of the learning policy and the optimal policy, which implies that the huge gap brought in by MACQL will severely affect the performance of the learned policy. Therefore, sampling OOD joint-actions from two or more agents may perform much worse than sampling just single OOD action from one agent. That is our reason behind the choice of sampling only OOD actions from one agent.

---

### Official Review · Reviewer_rt2G · 2023-07-23

**Soundness:** 3 good
**Presentation:** 3 good
**Contribution:** 3 good
**Rating:** 4
**Confidence:** 4

**Summary:**


This paper addresses challenges in Offline Multi-Agent Reinforcement Learning (MARL), which suffers from severe distribution shift issues and high dimensionality. To overcome these problems, the authors propose a novel MARL algorithm, CounterFactual Conservative Q-Learning (CFCQL). Unlike conventional methods that treat all agents as a single high-dimensional entity, CFCQL separately computes conservative regularization for each agent in a counterfactual manner and linearly combines them for an overall conservative value estimation. The authors demonstrate that CFCQL maintains the underestimation property and performance guarantee similar to single-agent methods but offers improved regularization and safe policy improvement bounds that are independent of the agent number. This method is thus advantageous, especially with large agent counts. Experimental validation on various environments and datasets shows that CFCQL outperforms existing methods significantly.



**Strengths:**

* The structure of the presentation is generally clear.
* The considered problem is interesting and of importance.





**Weaknesses:**

* The motivation for the algorithm is unclear, e.g. why considering counterfactual is helpful for offline MARL?
* The theoretical analysis is not well elaborated in context. E.g. the implications of theorems can be better elaborated.
* Some terms are not clearly defined, e.g. “PI”, “e_i” in Eq.(8).
* How algorithm 1  works with continuous or discrete action space is not clear.


**Questions:**

See weakness above.



**Limitations:**

* The presentation needs to be improved in terms of clarity, and rigor.

* The method is still employing centralized training, which limits its scalability.

---

> ### Author Rebuttal · Authors · 2023-08-08
>
> Thank you for your constructive feedback!
>
> ●**Q1: The motivation for the algorithm is unclear, e.g., why considering counterfactual is helpful for offline MARL?**
>
> We apologize for any confusion resulting from the unclear statement in our paper.
>
> The motivation for this study is rooted in the observation that directly transferring CQL into Multi-Agent settings, wherein the joint action is penalized, leads to an exponential increase in pessimism as the number of agents increases. Consequently, this results in an overly pessimistic value function.
>
> To mitigate this issue, it is a potential solution to separately penalize each agent's out-of-distribution (OOD) actions, then the penalty term will grow linearly instead of exponentially. This idea aligns perfectly with the counterfactual approach in MARL[1]. Specifically, counterfactual approaches aim to obtain each agent's counterfactual baseline by marginalizing out the agent's action while keeping the other agents' actions fixed to address the challenges associated with multi-agent credit assignment.
>
> In the context of our study, we successfully adapted the counterfactual method in offline multi-agent settings. This is achieved through the separate penalization of each agent's OOD actions, while holding the other agents' actions still sampled from datasets. Theoretically, this adaptation holds promise in alleviating the out-of-distribution problem with reduced pessimism. Moreover, the empirical results from our experiments serve to further validate the effectiveness of the counterfactual method in the domain of Offline Multi-Agent RL.
>
> ●**Q2: The theoretical analysis is not well elaborated in context, e.g., the implications of theorems can be better elaborated.**
>
> In this paper, we aim to theoretically analyze the pessimism of CFCQL and highlight its advantages over MACQL. In Theory 4.1, we rigorously demonstrate that our proposed updating rule, as represented by Eq. 4, leads to a lower bound of the value function. This finding validates the pessimistic property of CFCQL, which is necessary for offline settings. Moving forward, in Theory 4.2, we delve into a comparative study of the scale differences between CFCQL and MACQL in terms of their degree of pessimism. Our analysis reveals that the degree of pessimism exhibited by MACQL surpasses that of CFCQL and that the scale difference between the two methods tends to increase with the number of agents. Building upon the findings from Theory 4.2, we engage in a comprehensive discussion regarding the performance bound and safe policy improvement guarantee of both methods in Theory 4.3 and Theory 4.4, which allows us to showcase the dominance of CFCQL in comparison to MACQL, highlighting the superiority of our proposed approach.
>
> ●**Q3: Some terms are not clearly defined, e.g., “PI”, “e_i” in Eq. (8).**
>
> We sincerely apologize for the lack of clarity in our paper and any resulting confusion it may have caused. Allow us to clarify the notation in question. In line 281, "PI" stands for "policy improvement," denoting the iterative progress of policy update in the context of Reinforcement Learning. Additionally, in Eq.8, the symbol “e_i” represents a one-hot vector, wherein the i-th element assumes a value of 1 while all other elements remain 0.
>
> ●**Q4: How algorithm 1 works with continuous or discrete action space is not clear.**
>
> We sincerely apologize for any confusion that may have arisen from our algorithm implementation. Allow us to clarify the algorithm implementation for both discrete and continuous action spaces in our work.
>
> For the discrete action space, we utilized the QMIX algorithm, a well-known Multi-Agent Reinforcement Learning (MARL) Q-Learning method specifically designed for discrete action spaces as our backbone algorithm. Upon QMIX, we implement our counterfactual pessimism by the updating rule in Eq. 7.
>
> For the continuous action space, we opted for the MADDPG algorithm, a MARL algorithm based on deterministic policy gradient, as our backbone algorithm. In this setting, in addition to the replacement of update rule for the Q function (as denoted by Eq. 7), we also adapted the update rules for the policy function $\pi$ (as represented by Eq. 10) to align with the nature of the continuous action space scenario.
>
> We hope that these clarifications suffice to remove any ambiguity surrounding our algorithm implementation.
>
> Reference:
>
> [1] Foerster J, Farquhar G, Afouras T, et al. Counterfactual multi-agent policy gradients[C]//Proceedings of the AAAI conference on artificial intelligence. 2018, 32(1).

---

> ### Comment · Reviewer_rt2G · 2023-08-16
> **Further concerns**
>
>
> Thank you for the authors' response. I echo the Reviewer dhyd's concerns regarding the writing quality, particularly the informal and inconsistent notations. Many methodological details are absent as well.
>
> After reviewing the authors' response, I have additional concerns:
>
> 1. The distinctions among $\bar{Q}$, $\hat{Q}$, and ${Q}$ are not explained.
> 2. Line 6, which mentions  “updating by $\hat{\mathcal{T}}{\hat{\theta}}$ and $\hat{\mathcal{T}} {\hat{\theta}}^{\pi_{\psi_t}}$ ”, is unclear. What's the exact difference between these two operators is missing.
> 3. The term "Ablation study" in Figure 3 seems misplaced; it appears to merely examine hyperparameters.
> 4. The rationale for introducing the parameter $\lambda$ in Eq (4) is unclear. Moreover, the definition of $\bar{\mathcal{E}}$ in Eq.(4) is omitted.
> 5. The term “counterfactual” is concerning to me as it seems to refer to a marginal distribution without $a^i$. The paper lacks a clear explanation of what "counterfactual Q-learning" means.
> 6. The reviewer did not find explanation of the relations between policy $\pi, \mu$ and $\beta$ in line 181. Why is the update of $\beta^{-i}$ not included in Algorithm 1, which is used in Eq.7?

---

> > ### Author Response · Authors · 2023-08-17
> > **Reply to the comment of Reviewer rt2G**
> >
> > Thank you for your thorough proofreading! We apologize for the oversight in explaining certain symbols, and we're committed to addressing this issue and ensuring their proper clarification. Your feedback is greatly appreciated, and we'll strive to make the necessary improvements.
> >
> > **1. The distinctions among $\bar{Q}$, $\hat{Q}$, and $Q$ are not explained. The definition of $\bar{\epsilon}$ in Eq.(4) is omitted.**
> >
> > In principle, "\hat" refers to the symbols involved in MACQL, like the empirical $Q$ function, the empirical Bellman operator, the empirical TD error, etc. And "\bar" refers to the corresponding symbols involved in CFCQL.
> >
> > Concretely, we employ $\hat{Q}$ to represent the empirical Q function after each iteration in MACQL, and the $\bar{Q}$ to represent the empirical  Q function after each iteration in CFCQL. Similarly, $\bar{\mathcal{E}}$ serves a comparable role, signifying the empirical TD error in the loss of CFCQL, analogous to the function of $\hat{\mathcal{E}}$ in MACQL.
> >
> >  Regarding $Q$, it usually represents a variable that need to be optimized (Eq. 1,2,3,4,7), or the actual $Q$ function obtained via exact policy evaluation(Theorem 4.1), distinguished from the empirical $Q$. We will give detailed explanation about the symbol in the revised manuscript.
> >
> > **2. What's the difference between the two operators in Algorithm 1, Line 6?**
> >
> > These are two Bellman operators suitable for discrete and continuous action space, respectively. The main difference is that $\mathcal{T}$ uses the IGM principle to find the actions with max $Q^{tot}$ in the next time step, and $\mathcal{T}^{\pi}$ uses a parameterized policy to output the actions of the next time step. For detailed explanation, please refer to Section 3.2 "Value Functions in MARL".
> >
> > **3. The term "Ablation study" in Figure 3 seems misplaced.**
> >
> > Thank you for bringing this to our attention. We have made the necessary correction, now stating "hyperparameters examination."
> >
> > **4. The rationale for introducing the parameter $\lambda$.**
> >
> > The rationale behind parameter $\lambda$ is rooted in the understanding that uniformly penalizing the out-of-distribution (OOD) actions of each agent might not be optimal, considering that the OOD degrees across agents may vary. It's important to note that Theorem 4.2 necessitates only $\sum_i \lambda_i = 1$. Consequently, we introduce parameter $\lambda$ to impose higher penalties to agents displaying more significant deviations from the dataset. This is achieved under the constraint $\sum_i \lambda_i = 1$, ensuring that value function overestimation is avoided.
> >
> > **5. A clear explanation of what "counterfactual Q-learning" means.**
> >
> > In MARL, counterfactual refers to considering the return of $a^i$ during the training process, taking into account the situations where other agents take different actions. Therefore, the marginal distribution of $Q$ value without  $a^i$ is obtained by taking the expectation of different actions of other agents，and it only measures the impact of action $a^i$ on the $Q$ value[1].  In our context, when we pessimistically evaluate the $Q$ value for agent $i$, we  keep the other agents' actions sampled from the dataset , which is equivalent to only considering the pessimistic impact of $a^i$ on $Q$ values.  Therefore, our approach shares similar principles with the counterfactual method, so we adopted this term to refer to our method.
> >
> > **6.The relations between $\pi$, $\mu$ and $\beta$.  Why is the update of $\beta^{-i}$ not included in Algorithm 1?**
> >
> > First, let us clarify the significance of $\beta$. $\beta$ denotes an implicit distribution over the dataset $D$, signifying that the dataset can be considered as being collected by executing the behavior policy $\beta$ in the environment. Consequently, $\beta$ remains fixed and can not be updated. On the other hand, $\mu$ primarily emerges within the theoretical analysis of policy evaluation, and it can denote any policy different from the behavior policy $\beta$. The notation $\pi$ pertains to the learning policy. In practical algorithms, we often opt for $\mu=\pi$ for simplicity's sake, which is also consistent with the treatment in CQL[2]. For ease of understanding, you can equate $\mu$ and $\pi$, treating them as the same – that is, referring to the learning policy. This clarification will be accentuated in the revised manuscript.
> >
> > Reference:
> >
> > [1] Foerster J, Farquhar G, Afouras T, et al. Counterfactual multi-agent policy gradients[C]//Proceedings of the AAAI conference on artificial intelligence. 2018, 32(1).
> >
> > [2] Kumar, Aviral, et al. "Conservative q-learning for offline reinforcement learning." Advances in Neural Information Processing Systems 33 (2020): 1179-1191.

---

### Author Rebuttal · Authors · 2023-08-08

●Q1: Explanation on current baselines and more baseline results.

We acknowledge that the baselines used in our paper (OMAR, ICQ, and MADTKD) are designed for **offline** MARL, not off-policy MARL. We recognize that our paper lacks sufficient comparison with single agent offline RL methods except TD3-BC, the results of which have been listed in the original paper, Table 3 and Appendix Table 2.

Furthermore, following the request of Reviewer dhyd, we conducted additional experiments and have now incorporated the results of IQL and AWR into our comparison on **all** datasets appeared in the paper. To ensure better agent cooperation in continuous action spaces, similar to CFCQL, we applied the CTDE paradigm to both IQL and AWR. This involved utilizing a centralized Q/Value function while maintaining decentralized policies. Additionally, to enhance the stability of AWR, we adopted the more robust actor-critic alternate, AWAC[1].

The results on MaMuJoCo are listed below:

|Datasets|CFCQL|TD3-BC|IQL|AWAC|
|----------|--------|---------|-----|-------|
|Random|**39.7±4.0**|7.4±0.0|7.4±0.0|7.3±0.0|
|Med-Rep|**59.5±8.2**|27.1±5.5|58.8±6.8|30.9±1.6|
|Medium|80.5±9.6|75.5±3.7|**81.3±3.7**|71.2±4.2|
|Expert|**118.5±4.9**|114.4±3.8|115.6±4.2|113.3±4.1|

The results on MPE are listed below:

|Maps|Datasets|CFCQL|TD3-BC|IQL|AWAC|
|------|----------|--------|---------|-----|-------|
|CN|Random|**62.2±8.1**|9.8±4.9|5.5±1.1|0.5±3.7|
||Med-Rep|**52.2±9.6**|15.4±5.6|10.8±4.5|2.7±3|
||Medium|**65.0±10.2**|29.3±4.8|28.2±3.9|25.7±4.1|
||Expert|**112±4**|108.3±3.3|103.7±2.5|103.3±3.5|
|PP|Random|**78.5±15.6**|5.7±3.5|1.3±1.6|0.2±1.0|
||Med-Rep|**71.1±6**|28.7±20.9|23.2±12|8.3±5.3|
||Medium|**68.5±21.8**|65.1±29.5|53.6±19.9|50.9±19.0|
||Expert|**118.2±13.1**|115.2±12.5|109.3±10.1|106.5±10.1|
|World|Random|**68±20.8**|2.8±5.5|2.9±4.0|-2.4±2.0|
||Med-Rep|**73.4±23.2**|17.4±8.1|41.5±9.5|8.9±5.1|
||Medium|**93.8±31.8**|73.4±9.3|70.5±15.3|63.9±14.2|
||Expert|**119.7±26.4**|110.3±21.3|107.8±17.7|107.6±15.6|

The results on SMAC are listed below. No results of TD3-BC are provided in SMAC with discrete action space, since it is only suitable for continuous action space.

| Map | Dataset | CFCQL| IQL | AWAC |
| ------- | --------- | ------- | --------- | ---------------- |
|2s3z|Medium| **0.40±0.10** |0.16±0.04|0.19±0.05 |
||Medium_Replay|**0.55±0.07**|0.33±0.06 | 0.39±0.05|
||Expert|**0.99±0.01**|0.98±0.03| 0.97±0.03|
||Mixed|**0.84±0.09**|0.19±0.04| 0.14±0.04|
|3s_vs_5z|Medium| **0.28±0.03** |0.20±0.05|0.19±0.03 |
||Medium_Replay|**0.12±0.04**|0.04±0.04 | 0.08±0.05|
||Expert|**0.99±0.01**|**0.99±0.01**|**0.99±0.02** |
||Mixed|**0.60±0.14**|0.20±0.06|0.18±0.03 |
|5m_vs_6m|Medium| **0.29±0.05** |0.25±0.02|0.22±0.04 |
||Medium_Replay|**0.22±0.06**|0.18±0.04 |0.18±0.04 |
||Expert|**0.84±0.03**|0.77±0.03|0.75±0.02 |
||Mixed|0.76±0.07 |0.76±0.06| **0.78±0.02**|
|6h_vs_8z|Medium| 0.41±0.04|0.40±0.05| **0.43±0.06**|
||Medium_Replay|**0.21±0.05**|0.17±0.03 | 0.14±0.04|
||Expert|**0.7±0.06**|0.67±0.03| 0.67±0.03|
||Mixed|**0.49±0.08** |0.36±0.05|0.35±0.06 |

We have observed that CFCQL consistently outperforms IQL and AWAC, particularly on the Random dataset. This outcome aligns with our expectations since both IQL and AWAC can be considered as variants of weighted behavior cloning methods, and their performance heavily depends on the quality of the dataset used for training. In contrast, CFCQL does not necessitate the learned policies to closely match the behavior policy, which enables it to excel on datasets with lower quality.

Reference:

[1] Nair, Ashvin, et al. "Awac: Accelerating online reinforcement learning with offline datasets." arXiv preprint arXiv:2006.09359 (2020).

---

### Decision · Program_Chairs · 2023-09-21

**Decision:**

Accept (poster)

**Comment:**

This paper generated a lot of discussion between the reviewers and authors, as well as some internal discussions between the AC and reviewers. Reviewers felt that some of the details of the data composition and corresponding empirical evidence raised some doubts on the efficacy of the algorithm, though it is unclear if this is a flaw: while I think better data compositions could have been utilized, using medium, medium-replay data collection as the authors did is a popular protocol in literature, and I am not aware of any such protocol being utilized with multi-agent systems.

The other concern from reviewers was the presentation -- I agree that the submitted manuscript is not the best in terms of defining terminology and consistency in notation, but I am assuming that the authors will fix this if the paper is accepted. I also think that the paper needs to do a better job at differentiating their contribution and arguing why they attain better performance compared to prior multi-agent offline RL works.

Overall, the paper is borderline, but given that the discussion addressed a number of comments about the paper and since the paper does not deviate from commonly-expected design choices, I am inclining towards accepting this paper. That said, I do think that multi-agent offline RL would benefit from improved benchmarking, and encourage the authors to contribute to such an effort, including open-sourcing and releasing their datasets and environments.